# An integrated IKOA-CNN-BiGRU-Attention framework with SHAP explainability for high-precision debris flow hazard prediction in the Nujiang river basin, China

Hao Yang[1], Tianlong Wang[2,3]*, Nikita Igorevich Fomin[1], Shuoting Xiao[1], Liang Liu[1]

**1** Institute of Civil Engineering and Architecture, Ural Federal University, Yekaterinburg, Russia, **2** Ocean College, Zhejiang University, Zhoushan, China, **3** School of Civil and Environmental Engineering, Nanyang Technological University, Singapore, Singapore

\* tianlong_wang@zju.edu.cn

## Abstract

Debris flows represent a persistent challenge for disaster prediction in mountainous regions due to their highly nonlinear and multivariate triggering mechanisms. This study proposes an explainable deep learning framework, the Improved Kepler Optimization Algorithm-Convolutional Neural Network-Bidirectional Gated Recurrent Unit-Attention (IKOA-CNN-BiGRU-Attention) model, for precise debris flow hazard prediction in the Yunnan section of the Nujiang River Basin, China. The model is developed and validated using data from 159 debris flow-prone gullies, integrating deep convolutional, recurrent, and attention-based architectures, with hyperparameters autonomously optimized by IKOA. Model explainability is enhanced using SHapley Additive exPlanations (SHAP), which quantify the influence of key factors. The IKOA-CNN-BiGRU-Attention framework consistently outperforms 13 benchmark models, achieving a root mean square error of $2.33 \times 10^{-6}$, mean absolute error of $1.51 \times 10^{-6}$, and mean absolute percentage error of 0.006%. The model maintains high stability across 50 repeated experiments, strong resilience to 20% input noise, and robust generalizability under five-fold cross-validation. Interpretability analysis identifies potential source energy and maximum 24-hour rainfall as primary determinants and uncovers a dual-threshold physical mechanism underlying debris flow initiation. These findings provide a quantitative basis for adaptive early warning and targeted risk mitigation, and establish a transferable framework for explainable geohazard prediction.

## 1 Introduction

Debris flows constitute a catastrophic geophysical phenomenon that epitomizes the complex interplay between gravitational forces, material rheology, and hydrological dynamics in mountainous environments. These rapid mass movements pose an

**Data availability statement:** The complete dataset supporting all findings in this study is fully and openly accessible without any restrictions on Zenodo. All data is contained in a single downloadable Excel file (19.6 kB): Wang, T. (2025). Debris Flow Hazard-Causing Factors Dataset for the Yunnan Section of the Nujiang River Basin, China [Data set]. Zenodo. https://doi.org/10.5281/zenodo.15050116.

**Funding:** This work was supported by the China Scholarship Council (No. 202406320358) and the 1st Doctoral Student Program of the Young Elite Scientists Sponsorship Program by CAST (no award number). The funders had no role in study design, data collection and analysis, decision to publish, or preparation of the manuscript.

**Competing interests:** The authors have declared that no competing interests exist.

escalating threat to human settlements, critical infrastructure, and ecological integrity globally, with disproportionate impacts on developing regions with expanding mountain populations [1]. The socioeconomic burden is substantial: between 2000 and 2020, debris flows accounted for approximately 17% of global landslide-related fatalities while generating annual economic losses exceeding 20 billion USD. Unlike other natural hazards with extended warning periods, debris flows manifest with minimal antecedent indicators and propagate at velocities reaching 10 m/s, severely constraining evacuation timeframes and necessitating high-precision predictive frameworks for effective risk reduction [2]. The multifactorial triggering mechanisms underlying these events, characterized by nonlinear threshold responses and complex factor interactions, present formidable challenges for contemporary hazard forecasting.

The epistemological evolution of debris flow prediction has traversed distinct methodological paradigms, each characterized by specific conceptual frameworks and analytical limitations. Initial approaches relied on empirical correlations derived from historical event databases, establishing phenomenological relationships for inundation area delineation [3] and precipitation threshold determination [4]. Such frameworks, while operationally accessible, frequently exhibited limited transferability beyond their calibration domains. Subsequent advances introduced physically-based numerical models that simulated the complete process chain from material mobilization through deposition, incorporating rheological principles and granular flow mechanics [5]. These sophisticated simulations demonstrably enhanced impact assessment capabilities [6] and runout predictions in topographically complex terrain. Nevertheless, their widespread implementation remains constrained by intensive parameterization requirements, computational demands, and fundamental uncertainties in constitutive relationships governing material behavior during extreme mobilization events.

Intermediate methodological frameworks emerged to address these epistemological limitations through probabilistic formulations and multi-criteria decision architectures. Statistical models employing Bayesian inference frameworks quantified conditional hazard probabilities under varying environmental scenarios, enabling the establishment of robust rainfall thresholds with explicit uncertainty quantification [7]. Concurrently, multi-attribute decision frameworks provided structured methodologies for integrating heterogeneous hazard factors [8], with recent innovations applying Analytic Hierarchy Process techniques to susceptibility mapping across diverse geological contexts [9] and multi-criteria analysis approaches to flood risk assessment in coastal regions [10]. Despite their theoretical elegance, these approaches frequently encounter fundamental constraints in high-dimensional feature space integration and often introduce systematic biases through subjective parameter weighting schemes that compromise prediction reliability.

The integration of machine learning (ML) into geohazard assessment represents a paradigmatic transformation in debris flow prediction, facilitating the identification of complex, nonlinear patterns within high-dimensional environmental datasets that remain inaccessible to conventional analytical approaches [11]. Contemporary

research has demonstrated the efficacy of artificial neural networks in characterizing rainfall-triggered debris flow probability distributions and advanced information-theoretic coupling methods for susceptibility assessment [12]. These computational frameworks excel in capturing intricate multivariate relationships without requiring explicit physical process formulation [13]. Furthermore, ensemble learning techniques enhance predictive stability and generalization capabilities across heterogeneous environmental conditions previously inaccessible to traditional modeling frameworks.

Despite their transformative potential, current ML applications to debris flow hazard assessment exhibit critical epistemological and methodological limitations that constrain their scientific impact and operational utility. First, model performance demonstrates an acute sensitivity to hyperparameter configurations, with suboptimal parameterization inducing overfitting phenomena that compromise generalization to novel geomorphological contexts [14]. Second, conventional optimization algorithms frequently converge prematurely to local extrema, resulting in suboptimal exploration of the high-dimensional parameter space [15]. Third, and most fundamentally, deep learning architectures manifest intrinsic opacity that creates a profound disconnect between predictive capacity and mechanistic understanding [16]. This explainability deficit, frequently characterized as the "black-box problem," represents a critical epistemological barrier in hazard forecasting applications. The black-box problem specifically refers to the inherent inability to trace how deep learning models transform input features into output predictions through their complex internal architectures. This opacity manifests through (1) algorithmic complexity, where numerous interconnected neurons and non-linear activation functions obscure input-output relationships; (2) latent feature representation, where models develop abstract internal representations that lack direct physical interpretation; and (3) stochastic learning behavior, where training procedures yield models whose internal configurations cannot be deterministically predicted. Contemporary neural networks, while achieving unprecedented predictive accuracy, typically obscure the contribution of specific variables to prediction outcomes, thereby inhibiting scientific elucidation of causative mechanisms and undermining the implementation of targeted mitigation strategies.

Three fundamental research gaps persist in contemporary literature that motivate this study: (1) the predominant prioritization of predictive accuracy over model explainability creates an artificial tension between computational performance and theoretical advancement; (2) current methodological frameworks overwhelmingly employ single-algorithm approaches that inadequately capture the complex spatiotemporal characteristics of debris flow triggering conditions; and (3) the systematic integration of advanced optimization techniques with explainable artificial intelligence frameworks remains largely unexplored in geohazard prediction contexts, particularly for multi-parameter debris flow risk assessment. Explainable Artificial Intelligence (XAI) methodologies, particularly the SHapley Additive exPlanations (SHAP) approach, offer promising solutions to these challenges by providing mathematically rigorous attribution mechanisms derived from cooperative game theory principles [17,18]. Despite preliminary applications identifying dominant controlling factors in debris flow research, comprehensive XAI integration within holistic debris flow assessment frameworks remains substantially underdeveloped [19].

Motivated by these research gaps and the potential of XAI approaches, this study aims to develop and validate an integrated deep learning framework that maximizes both predictive accuracy and scientific explainability for debris flow hazard assessment. The specific research objectives include: (1) identifying optimal architectural configurations for capturing complex spatiotemporal characteristics of debris flow triggering factors; (2) quantifying the relative contribution of environmental, geological, and meteorological variables; (3) establishing a theoretical framework connecting physical processes to prediction outcomes through explainable AI; and (4) determining quantitative thresholds governing debris flow risk to support evidence-based intervention strategies.

To address these objectives, this study implements an IKOA-CNN-BiGRU-Attention framework that integrates physics-inspired optimization algorithms with deep learning architectures and explainability mechanisms. This framework combines convolutional networks for spatial feature extraction, bidirectional recurrent units for temporal dynamics modeling, and attention mechanisms for feature importance weighting, specifically designed to capture debris flow triggering mechanisms while providing SHAP-based scientific insights. The framework advances debris flow hazard prediction

methodologically, theoretically, analytically, and practically, offering actionable insights for precision disaster risk management. The overall workflow of the study is illustrated in Fig 1.

## 2 Methodology

### 2.1 Comprehensive weighting strategy

Precise determination of influencing factor weights constitutes a fundamental challenge in debris flow hazard assessment. This study proposes a novel game theory-based comprehensive weighting framework that integrates three complementary methodologies: the coefficient of variation method (CVM), entropy weight method (EWM), and CRITIC method [20]–[22]. Each methodology captures distinct aspects of factor importance: data dispersion, information entropy, and inter-factor correlations, respectively. The proposed framework applies cooperative game theory principles to optimize the

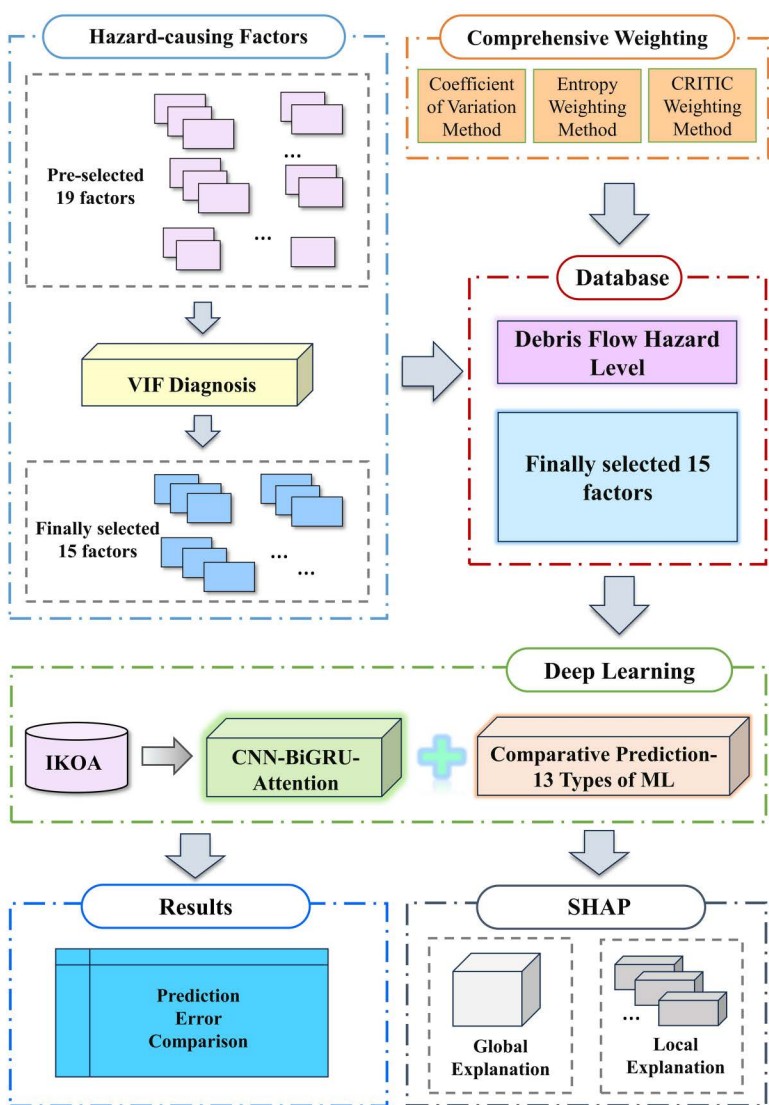

**Fig 1. Flow chart of this study.**

collective strengths of these methods, conceptualizing each as a strategic element in a cooperative game to determine optimal weight distribution. The complete framework is illustrated in Fig 2.

## 2.2 Improved parameter adjustment strategy for swarm intelligence algorithms

**2.2.1 The KOA algorithm.** The Kepler Optimization Algorithm (KOA), introduced by Mohamed Abdel-Basset in 2023, serves as the foundation for model optimization in this study. KOA represents a physics-based metaheuristic algorithm inspired by Kepler's three laws of planetary motion [23]. This algorithm conceptualizes the Sun as the optimal solution and planets as candidate solutions, with planetary motion mechanisms driving the evaluation of solution quality. Compared to established optimization algorithms such as Grey Wolf Optimizer (GWO) and Sparrow Search Algorithm (SSA), KOA exhibits enhanced exploration capabilities within the solution space through innovative search strategies, enabling effective avoidance of local optima entrapment while ensuring stability and robustness in hyperparameter optimization for complex models. From a mathematical perspective, KOA operates through seven sequential steps:

**Step 1:** Initialization process. The algorithm begins with a random generation of candidate solutions representing planets moving within the search space. Each planet possesses a position and velocity representing potential optimization solutions. Each planetary position is formally defined as shown in Equation (1). In Equation (1), $X_i$ represents the $i^{th}$ candidate solution in the model; $d$ represents the number of dimensions; $N$ indicates the number of candidate solutions; $X^j_{i,up}$

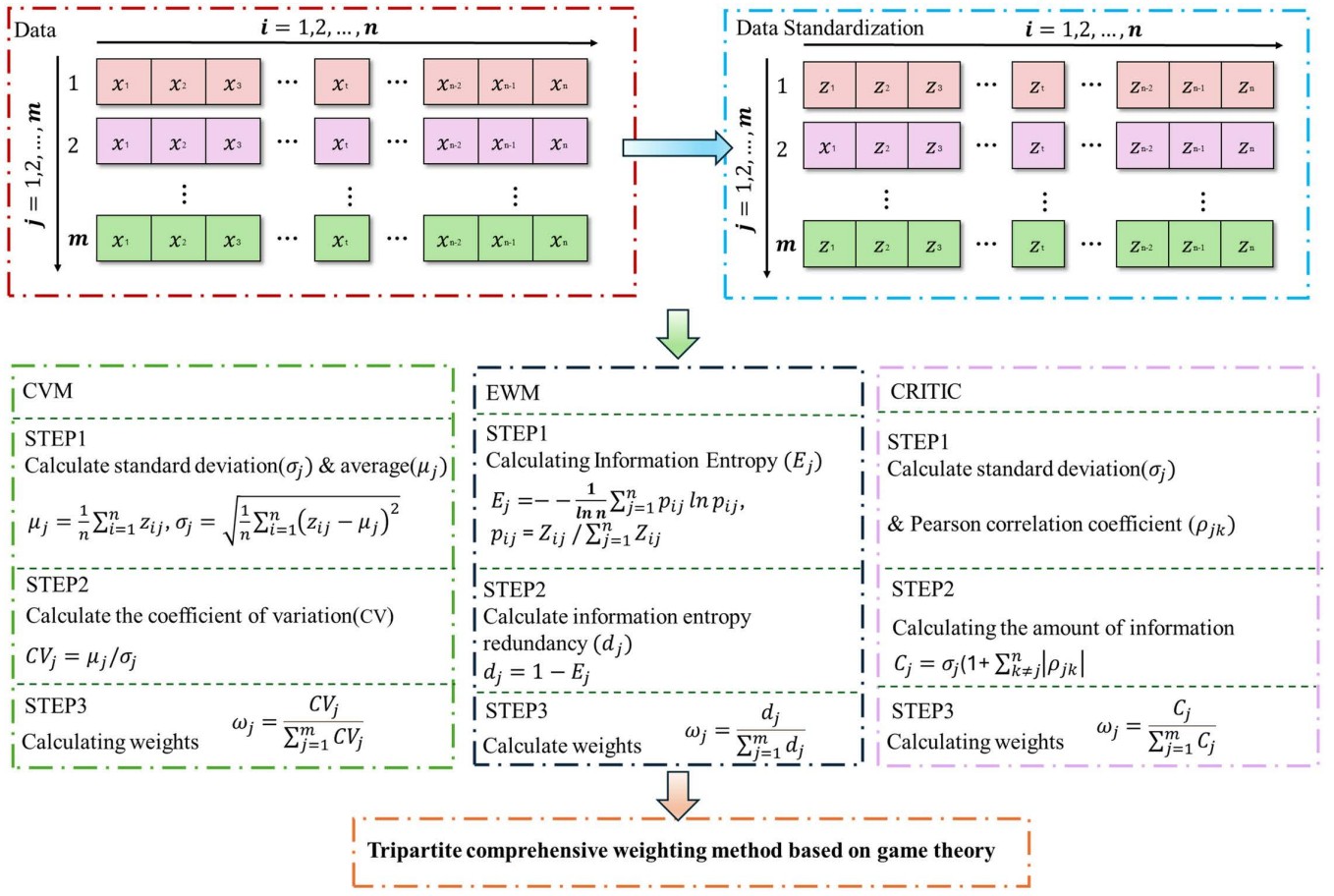

**Fig 2. Framework of the game theory-based comprehensive weighting strategy.**

and $X_{i,low}^j$ determine the boundary range of the search space, with $rand_{[0,1]}$ randomly selected values between $0$ and $1$. The eccentricity $e_i$ and period $T_i$ are initialized according to Equations (2) and (3). In Equations (2) – (3), $r$ is a random number based on a normal distribution.

$$X_i^j = X_{i,low}^j + rand_{[0,1]} \times \left( X_{i,up}^j - X_{i,low}^j \right), \begin{cases} i = 1, 2, \ldots, N \\ j = 1, 2, \ldots, d \end{cases}$$

(1)

$$e_i = rand_{[0,1]}, i = 1, 2, \ldots, N$$

(2)

$$T_i = |r|, i = 1, 2, \ldots, N$$

(3)

**Step 2:** Defining the gravitational force ($F$). Planets experience varying velocities at different orbital positions due to differential gravitational pull from the Sun along elliptical planetary orbits. The gravitational force experienced by a planet is formulated as in Equation (4), while the Euclidean distance is represented in Equation (5).

$$F_{gi} = e_i \times \mu(t) \times \frac{\overline{M}_s \times \overline{m}_i}{\overline{R}^2 + \varepsilon} + r_1$$

(4)

$$R_i(t) = \|X_s(t) + X_i(t)\|_2 = \sqrt{\sum_{j=1}^{d} \left( X_{Sj}(t) - X_{ij}(t) \right)^2}$$

(5)

**Step 3:** Calculating object velocity. Based on gravitational force and Sun-planet distance, the velocity of a planet at any position on its elliptical orbit is expressed as Equation (6).

$$V_i(t) = \begin{cases} 1 \times (2r_4 \vec{X}_i - \vec{X}_b) + (1 - R_{i-norm}(t)) \times \tau \\ \times \vec{U}_1 \times \vec{r}_5 \left( \vec{X}_{i,up} - \vec{X}_{i,low} \right), if R_{i-norm}(t) \leq 0.5 \\ r_4 \times \delta \times \left( \vec{X}_a - \vec{x}_i \right) + (1 - R_{i-norm}(t)) \times \tau \times \vec{U}_2 \\ \times \vec{r}_5 \times \left( r_3 \vec{X}_{i,up} - \vec{X}_{i,low} \right), Else \end{cases}$$

(6)

**Step 4:** Escaping from local optima. The parameter $\pi$ in the velocity equation modulates search direction, simulating varying orbital directions of planets within the Solar System. This mechanism prevents convergence on local optima, as shown in Equation (7).

$$\tau = \begin{cases} 1, if r_4 \leq 0.5 \\ -1, Else \end{cases}$$

(7)

**Step 5:** Updating object positions. Planetary motion occurs in two distinct phases: approaching and receding from the Sun. These correspond to the exploitation and exploration phases respectively. During exploration, the algorithm discovers new candidate solutions. During exploitation, solutions near the optimum are refined, as shown in Equation (8).

$$\vec{X}_i(t+1) = \vec{X}_i(t) + \tau \times V_i(t) + (F_{gi}(t) + |r|) \times \vec{U} \times (\vec{X}_s(t) - \vec{X}_i(t))$$

(8)

**Step 6:** Updating Sun-planet distance. Manipulation of parameter h enhances exploration and exploitation capabilities, as shown in Equation (9).

$$\vec{X}_i(t+1) = \vec{X}_i(t) \times \vec{U}_1 + (1 - \vec{U}_1) \times \left( \frac{\frac{\vec{X}_i(t) + \vec{X}_s + \vec{X}_a(t)}{3.0} + h}{\times \left( \frac{\vec{X}_i(t) + \vec{X}_s + \vec{X}_a(t)}{3.0} - \vec{X}_b(t) \right)} \right)$$

(9)

**Step 7:** Elitism mechanism. The algorithm evaluates fitness values between current and newly calculated positions, retaining superior solutions while discarding inferior ones, as shown in Equation (10).

$$\vec{X}_{i,new}(t+1) = \begin{cases} \vec{X}_i(t+1), if\, f(\vec{X}_i(t+1)) \leq f(\vec{X}_i(t)) \\ \vec{X}_i(t), Else \end{cases}$$

(10)

**2.2.2 The IKOA algorithm.** Despite the promising capabilities of physics-inspired metaheuristics, optimization algorithms frequently encounter fundamental challenges including premature convergence, local optima entrapment, and suboptimal exploration-exploitation balance. To overcome these intrinsic limitations in the standard KOA framework, this study proposes a systematically enhanced variant, the Improved Kepler Optimization Algorithm (IKOA), incorporating three complementary methodological refinements: (1) Chebyshev mapping for ergodic population initialization, (2) golden sine operator for solution space adaptation, and (3) dynamic weight coefficient for balanced search dynamics. This integrative approach ensures exceptional parameter optimization robustness across complex hypersurfaces. The complete algorithmic architecture is illustrated in Fig 3.

1. Improved strategy for population initialization based on Chebyshev mapping

Chaos mapping represents an uncertain and random methodology within nonlinear dynamic systems. In the process of initializing position updates, chaotic variables replace the random variables traditionally used in intelligent algorithm processes, thereby offering a broader search range in the solution space compared to probability-based random search strategies. Consequently, employing chaos mapping for population initialization in the KOA enhances the diversity of the population and the convergence speed of the optimization algorithm. This study utilizes the Chebyshev chaos mapping method [24], with its mathematical description presented in Equation (11), where $k$ denotes the order.

$$x_{n+1} = \cos(k\, arccos x_n), x_n \in [-1, 1]$$

(11)

2. Rolling position update strategy based on golden sine algorithm

The Golden Sine Algorithm (Gold-SA), proposed by Tanyildizi et al., is a novel metaheuristic algorithm [25]. Gold-SA differs from swarm intelligence algorithms by focusing on enhancing search efficiency and optimization precision in the later stages through the gradual narrowing of the current solution space. This means that the current search space continuously converges towards the optimal target value. In the traditional KOA algorithm, the efficiency of local and global searches by planetary individuals is relatively low, making the algorithm more susceptible to becoming trapped in local optima as it progresses. By incorporating the golden sine operator into Step 5, and leveraging the properties of the golden ratio coefficient to reduce the solution space, the overall performance of the algorithm is significantly improved. The position update formula with the added golden sine operator is presented in Equation (12). In Equation (15), $R_1 \in [0, 2\pi]$ is a random number determining the distance of the individual's movement in the next iteration, $R_2 \in [0, \pi]$ is a random number determining the direction of the next movement, and $x_1 = -\pi + (1-\tau)*2\pi$, $x_2 = -\pi + \tau*2\pi$ are coefficients derived from the golden section ratio $\tau = (\sqrt{5}-1)/2$, This ratio narrows the algorithm's search space and guides individuals gradually towards the optimum.

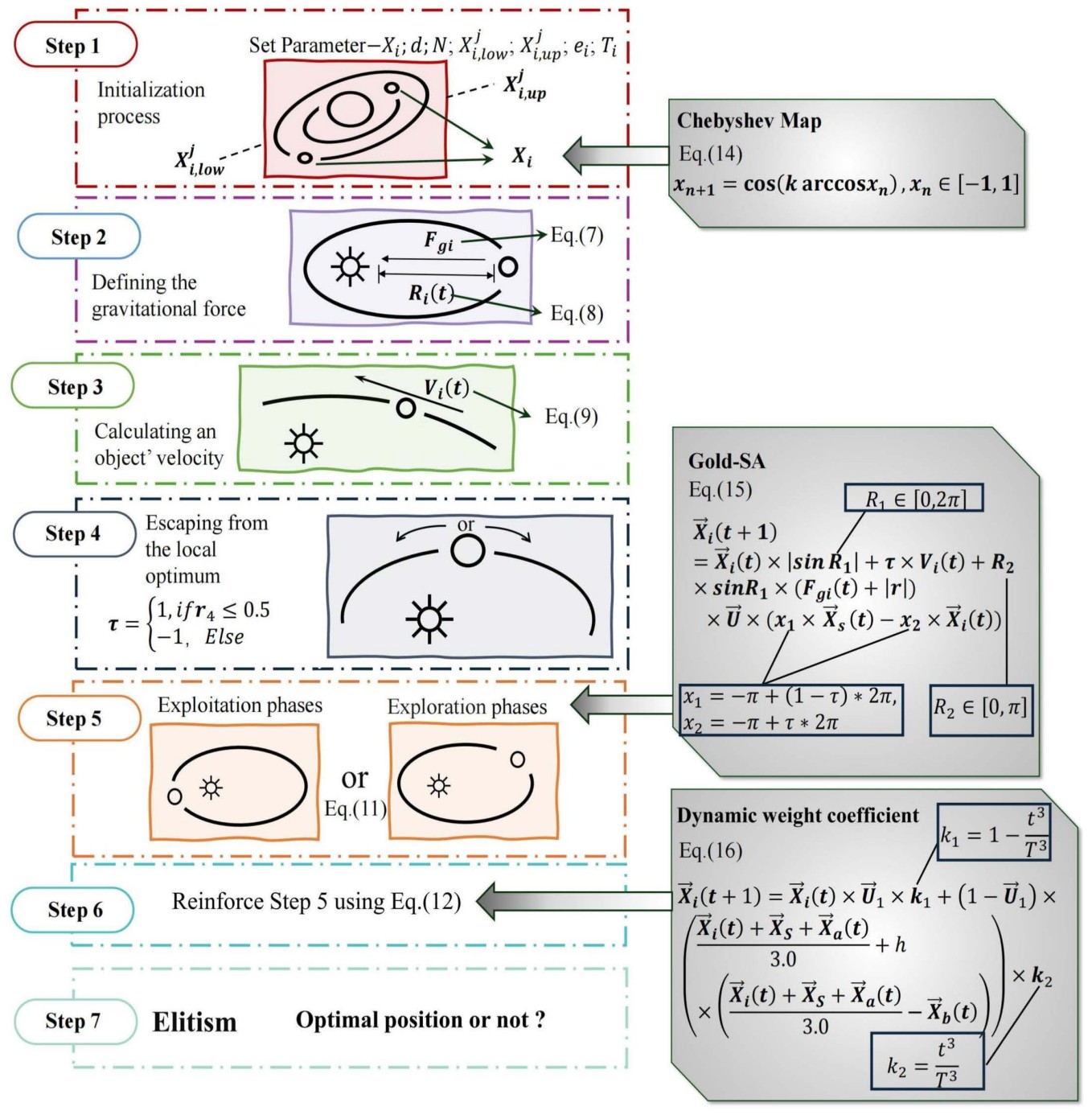

**Fig 3. Flow chart of the Improved Kepler Orbital Algorithm (IKOA).**

$$\vec{X}_i(t+1) = \vec{X}_i(t) \times \left| \sin R_1 \right| + \tau \times V_i(t) + R_2 \times sinR_1 \times (F_{gi}(t) + \left| \, r \right|) \times \vec{U} \times (x_1 \times \vec{X}_s(t) - x_2 \times \vec{X}_i(t)) \tag{12}$$

### 3. Position update strategy based on dynamic weight coefficient

To address the issue of insufficient global exploration capability during the position updates for planets close to the Sun in Step 6, a dynamic weight coefficient based on the number of iterations is introduced in Equation (9). This addition provides a balanced capability for global exploration and local exploitation of the algorithm, as detailed in the improved formula shown in Equation (13). In Equation (13), $k_1 = 1 - t^3/T^3$, $k_2 = t^3/T^3$. Within Equation (13), $k_1$ is larger in the early stages of iteration, enhancing the planets' early exploration ability, while $k_2$ gradually increases in the later stages, ensuring the planets' local exploitation capability in the latter phase.

$$\vec{X}_i(t+1) = \vec{X}_i(t) \times \vec{U}_1 \times k_1 + (1 - \vec{U}_1) \times \left( \begin{array}{c} \frac{\vec{X}_i(t) + \vec{X}_s + \vec{X}_a(t)}{3.0} + h \\ \times \left( \frac{\vec{X}_i(t) + \vec{X}_s + \vec{X}_a(t)}{3.0} - \vec{X}_b(t) \right) \end{array} \right) \times k_2 \tag{13}$$

**2.2.3 Benchmark function test.** To evaluate IKOA's optimization capabilities, this study implemented a benchmark analysis using nine standard test functions from the CEC 2005 benchmark suite [26]. These functions provide standardized performance metrics across multiple dimensions: unimodal functions $f_1$-$f_5$ assess local exploitation efficiency; multimodal functions $f_6$-$f_7$ examine global exploration capabilities; and fixed-dimension multimodal functions $f_8$-$f_9$ evaluate performance in complex search spaces. Table 1 presents the mathematical formulations and characteristics of each function. Experimental dimensionality was set at 30 for functions $f_1$-$f_7$ and 2 for functions $f_8$-$f_9$, following standard protocols. The comparative analysis included IKOA, original KOA, and three contemporary algorithms: Grey Wolf Optimizer (GWO), Sparrow Search Algorithm (SSA), and Whale Optimization Algorithm (WOA) [27–29]. Each function underwent 30 independent optimization trials, with convergence trajectories illustrated in Fig 4. The analysis demonstrates IKOA's superior performance across optimization metrics. IKOA consistently exhibits accelerated convergence compared to reference algorithms, achieving superior early-stage exploration and late-stage exploitation precision while maintaining an exceptional balance between search diversification and intensification. These qualities establish IKOA's theoretical

**Table 1. Benchmark function information.**

| Function | Dim | Space | Min |
|---|---|---|---|
| $f_1(x) = \sum_{i=1}^{n} x_i^2$ | 30 | $[-100,100]$ | 0 |
| $f_2(x) = \sum_{i=1}^{n} \left| x_i \right| + \prod_{i=1}^{n} x_i$ | 30 | $[-10,10]$ | 0 |
| $f_3(x) = \sum_{i=1}^{n} \left( \sum_{j=1}^{i} x_j^2 \right)$ | 30 | $[-100,100]$ | 0 |
| $f_4(x) = max\left\{ \left| x_i \right|, 1 \leq i \leq n \right\}$ | 30 | $[-100,100]$ | 0 |
| $f_5(x) = \sum_{i=1}^{n} -x_i sin(\sqrt{\left| x_i \right|})$ | 30 | $[-500,500]$ | 0 |
| $f_6(x) = 418.9829n - \sum_{i=1}^{n} x_i \sin(\sqrt{\left| x_i \right|})$ | 30 | $[-500,500]$ | 0 |
| $f_7(x) = \sin^2(\pi\omega_1) + \sum_{i=1}^{d-1} (\omega_1 - 1)^2 [1 + 10\sin^2(\pi\omega_1 + 1)]$ $(\omega_d - 1)^2] + [1 + \sin^2(2\pi\omega_d)], \omega_i = 1 + \frac{(x_i - 1)}{4}, \forall_i = 1, 2\ldots, d$ | 30 | $[-10,10]$ | 0 |
| $f_8(x) = \sin^2(3\pi x_1) + (x_1 - 1)^2 [1 + \sin^2(3\pi x_2)]$ $+ (x_2 - 1)^2 [1 + \sin^2(2\pi x_2)]$ | 2 | $[-10,10]$ | 0 |
| $f_9(x) = (1.5 - x_1 + x_1 x_2)^2 + (2.625 - x_1 + x_1 x_2^3)^2$ | 2 | $[-4.5,4.5]$ | 0 |

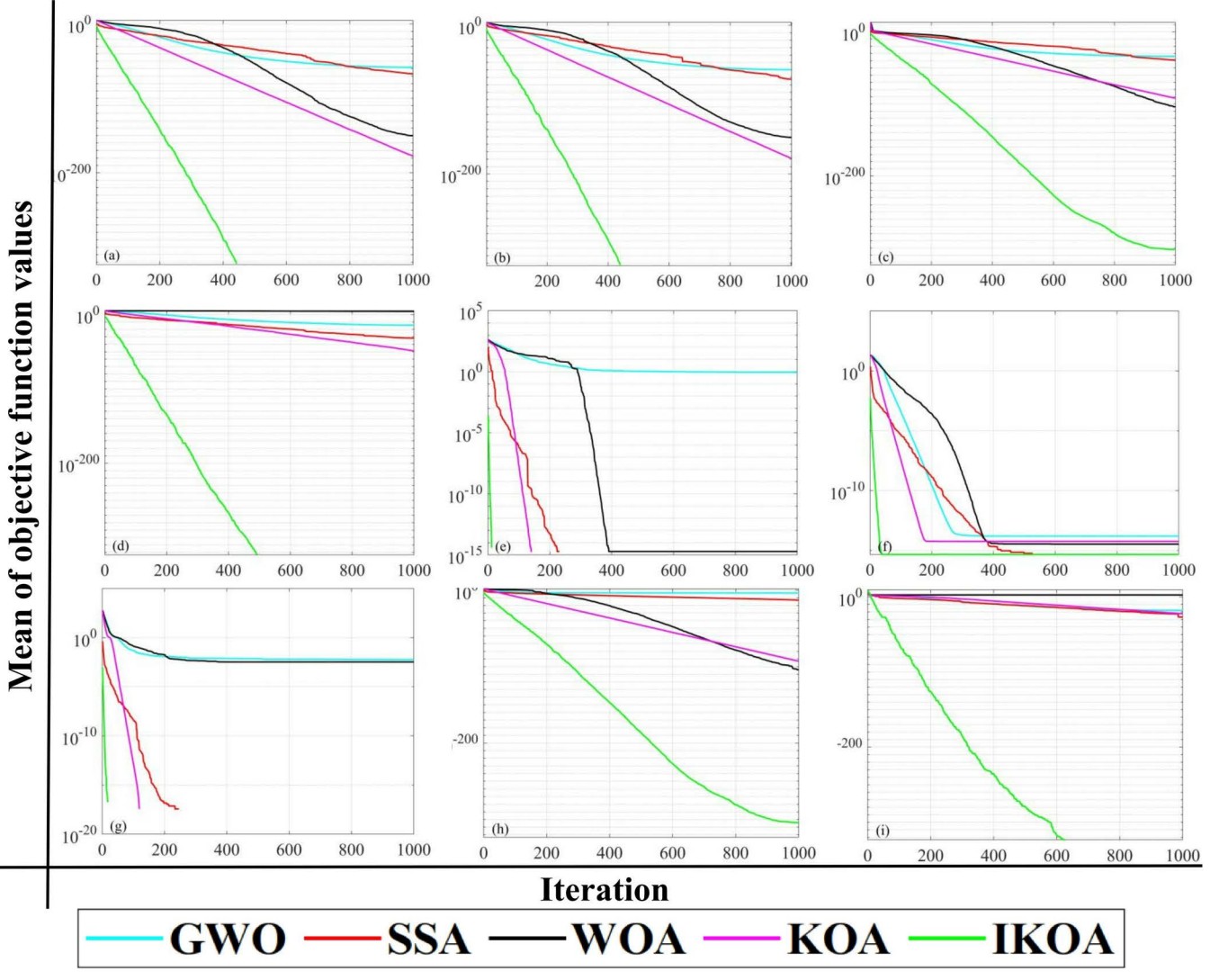

**Fig 4. The convergence curve of each optimization algorithm under the test function.**

foundation for application in debris flow hazard assessment modeling, where its enhanced optimization capabilities directly translate to improved predictive performance.

## 2.3. The IKOA-CNN-BiGRU-Attention model

This study presents a novel hierarchical deep learning architecture for debris flow hazard assessment that integrates optimization algorithms with specialized neural network components. The IKOA-CNN-BiGRU-Attention framework, illustrated in Fig 5, implements a cascaded processing pipeline for multidimensional geophysical data. The CNN module [30] employs convolutional operations to extract spatially invariant features across multiple abstraction levels, identifying complex topographical signatures that serve as precursors to debris flow events. The BiGRU component [31] extends representation capacity through bidirectional recurrent processing, capturing non-linear dependencies in sequential geophysical processes by propagating information in forward and reverse temporal directions. The Attention mechanism [32]

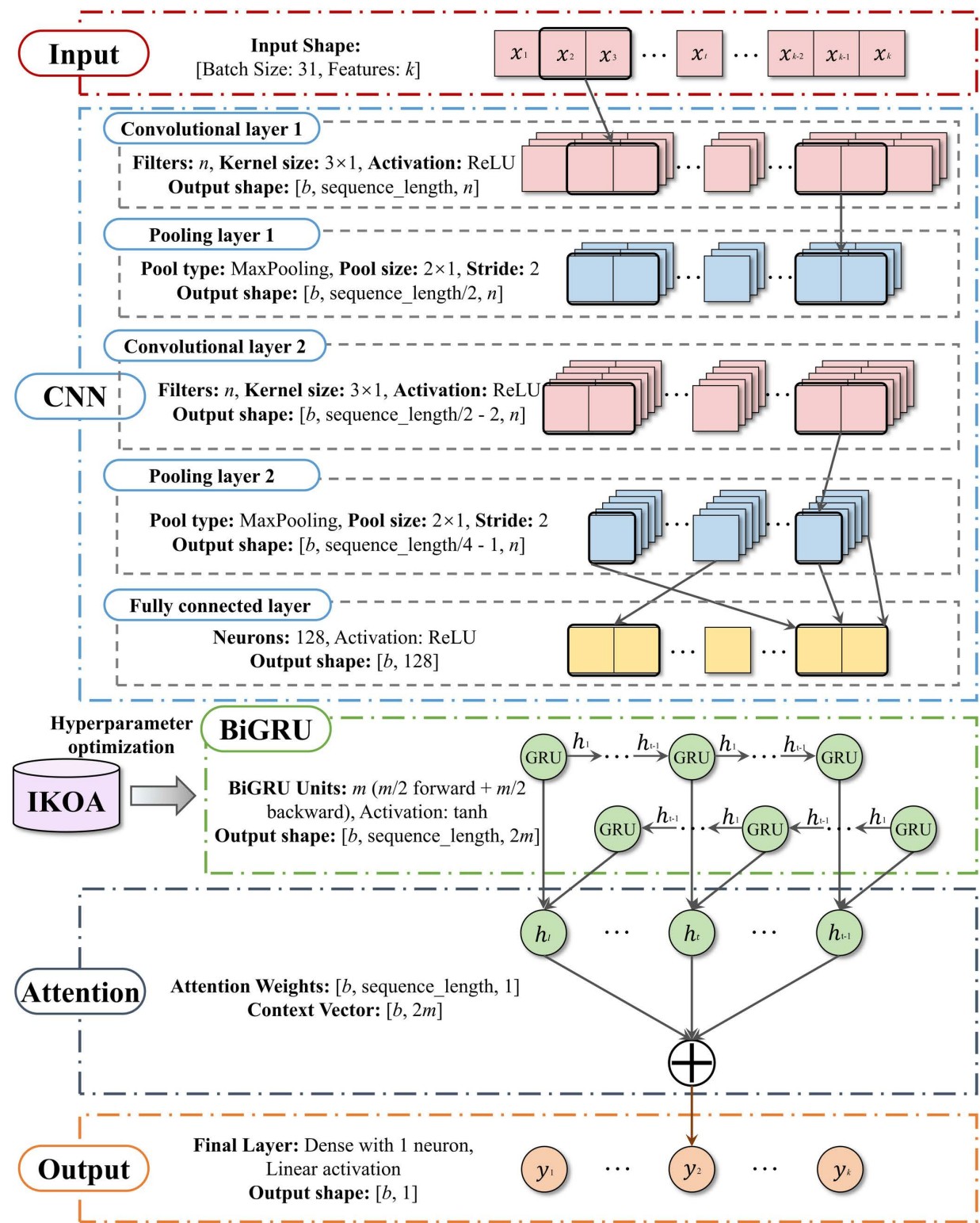

**Fig 5. Architectural framework of the integrated IKOA-CNN-BiGRU-Attention model.**

implements adaptive computational focus through learned importance weighting, dynamically modulating feature contributions based on contextual relevance. A distinctive contribution is the meta-optimization approach implemented through IKOA, which systematically refines four foundational hyperparameters: Learning Rate, GRU Units $m$, CNN Filters $n$, and Batch Size $b$. These parameters critically influence model performance, balancing expressiveness, computational efficiency, and generalization capability. The implementation proceeds through four systematic phases:

**Step 1:** Data Preprocessing. Statistical standardization establishes numerical stability while K-Nearest Neighbors imputation preserves data integrity through the reconstruction of missing values.

**Step 2:** Model Initialization. The preprocessed dataset undergoes chronological partitioning with 70% allocated for parameter estimation and feature learning.

**Step 3:** Hyperparameter Optimization. IKOA explores the hyperparameter space to identify optimal architectural configurations through iterative performance evaluation.

**Step 4:** Model Deployment. The optimized architecture is applied to the remaining 30% for validation and hazard prediction generation.

## 3 Materials

The study area encompasses the Yunnan section of the Nujiang River Basin in China (Fig 6), spanning approximately 33,500 km² and constituting 8.7% of Yunnan Province's total area. This geographically significant region extends 650 km along the Nujiang (Salween) River, bordered by the Lancang River Basin to the east and Myanmar to the west, spanning from the southern margins of the Tibetan Plateau to the northern extent of the Yunnan-Guizhou Plateau. The study area is situated between 24°07' to 28°23' N latitude and 98°07' to 100°00' E longitude. The geomorphology exhibits extreme topographic heterogeneity characterized by dramatic alpine gorges and precipitous mountainous terrain. The landscape features a pronounced north-south elevation gradient, intersected by the Hengduan Mountain Range. The area contains one of the most significant geomorphological features in the region, the Nujiang Grand Canyon, which ranks among the world's deepest canyon systems. This exceptional topographic amplitude creates optimal conditions for gravitational mass movements, particularly debris flows with substantial kinetic energy and destructive potential.

The region's climate is classified as humid subtropical monsoon, characterized by abundant yet spatially heterogeneous precipitation. Annual rainfall demonstrates significant variability, with maximum precipitation reaching approximately 1500 mm in higher elevation zones and diminishing to below 1000 mm in lower elevation areas. Temporal distribution exhibits marked seasonality, with over 51% of annual precipitation concentrated during the summer monsoon period [27]. This concentrated rainfall pattern constitutes the primary triggering mechanism for debris flow mobilization. The geological framework displays exceptional complexity, dominated by metamorphic complexes and sedimentary formations intersected by numerous fault systems. This intricate structural geology, combined with intense neotectonic activity, has produced extensive zones of fractured and weathered lithological material that serves as source material for debris flows. The synergistic interaction between fractured bedrock, steep topographic gradients, and concentrated seasonal precipitation creates ideal conditions for debris flow initiation and propagation. These distinctive geoenvironmental characteristics contribute to the elevated frequency and magnitude of geohazard events throughout the region. The prevalence of these hazards necessitates systematic investigation for effective disaster risk reduction strategies and sustainable regional development initiatives. This research utilizes comprehensive data from 159 debris flow prone gullies distributed throughout the Nujiang River Basin, applying advanced deep learning methodologies to analyze and predict debris flow hazard intensities across diverse geomorphological contexts.

Elevation data are based on SRTM DEM (Shuttle Radar Topography Mission, NASA/USGS). Administrative boundaries and rivers are from Natural Earth (public domain). Map generated by the authors.

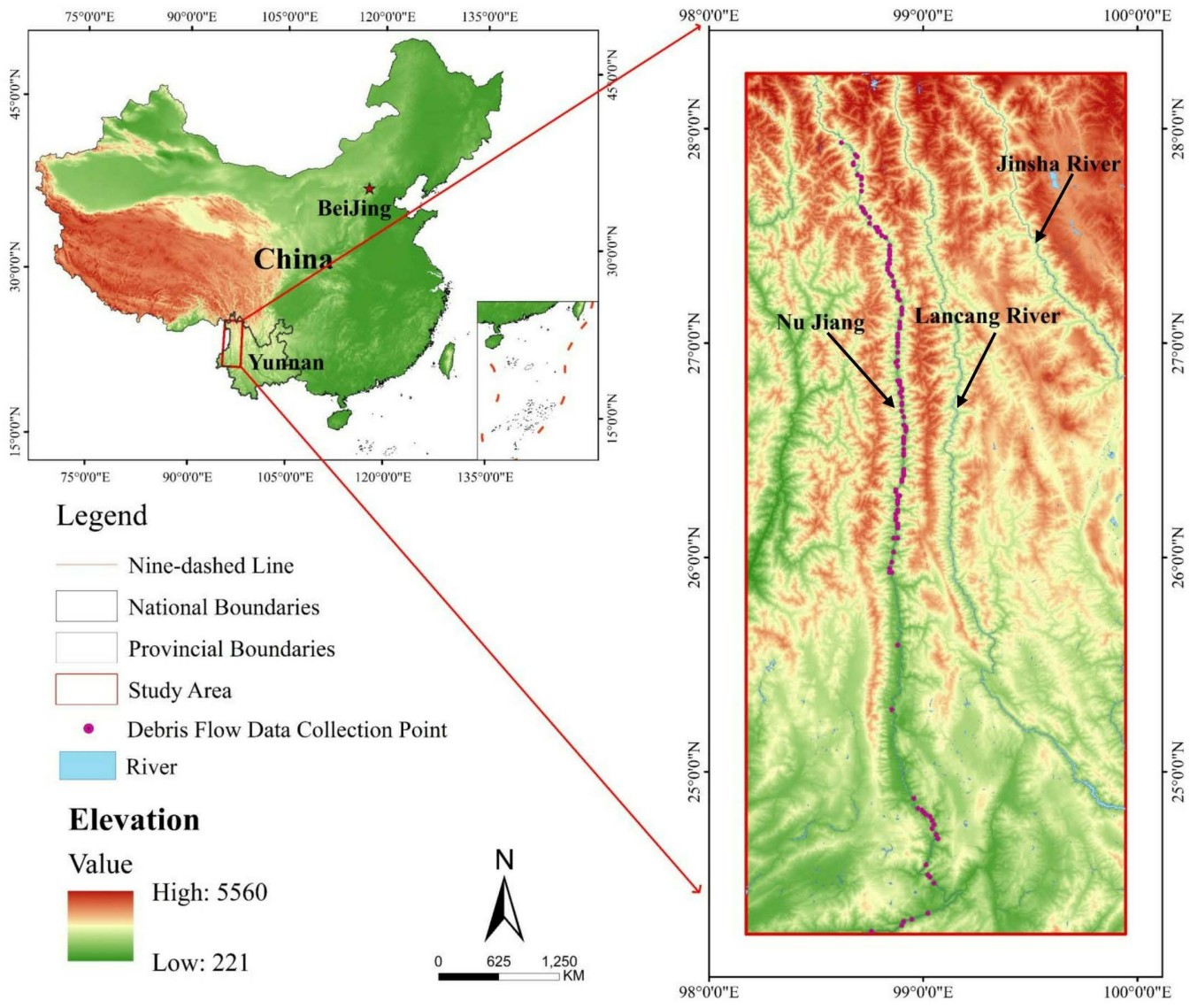

**Fig 6. Geographical distribution of the study area.**

## 4 Hazard-causing factors and weighting calculation

### 4.1 Selection of hazard-causing factors

This investigation established a comprehensive parametric foundation for debris flow susceptibility analysis through systematic integration of multidisciplinary data sources, including extensive literature synthesis [28], satellite remote sensing explanation, field-based geomorphological investigations, and archival geological and hydrological documentation. Advanced geospatial technologies facilitated rigorous extraction of topographical variables from 30m-resolution SRTM DEM data, while meteorological parameters underwent spatial interpolation through geostatistical inverse distance weighting algorithms calibrated with observational data from meteorological stations distributed throughout the study domain.

The extracted factors for the study area include: maximum 24-hour rainfall (X1), annual average rainfall (X2), annual average temperature (X3), annual average wind speed (X4), annual average vapor pressure (X5), annual average relative humidity (X6), potential source energy (X7), catchment area (X8), average gradient of the main channel (X9), length ratio of the sediment supply section (X10), length of the main channel (X11), sinuosity of the main channel (X12), drainage density of the catchment (X13), valley profile roundness (X14), vegetation cover rate (X15), average slope (X16), maximum relative elevation difference within the catchment (X17), lithological impact (X18), and dynamic storage capacity (X19). A total of 19 factors were selected, sequentially numbered as X1 to X19. In line with the principles of systematicity, completeness, effectiveness, and comparability in the selection of evaluation indicators, and considering the accessibility and operability of the data, the debris flow outbreak system in the study area was divided into three subsystems: the hydro-meteorological subsystem, material source subsystem, and geographic subsystem, encompassing the aforementioned 19 factors (X1-X19). The rationale for selecting each factor is detailed in Table 2.

## 4.2 Diagnosis of multicollinearity

Statistical independence among predictor variables constitutes a critical prerequisite for robust hazard modeling. The research implemented Variance Inflation Factor (VIF) analysis to systematically diagnose multicollinearity and optimize variable selection. VIF quantifies variance inflation of regression coefficients due to linear dependencies, calculated as $1/(1-R^2)$, where $R^2$ represents the coefficient when regressing a variable against all other predictors. While conventional guidelines suggest a VIF threshold of 10, the current investigation adopted a more conservative threshold of 5 to ensure model stability and parameter precision for complex environmental systems [51]. Multicollinearity diagnosis identified four factors exceeding this threshold: X10 (length ratio of the sediment supply section: 7.83), X14 (valley profile roundness: 5.62), X17 (maximum relative elevation difference: 8.14), and X19 (dynamic storage capacity: 6.95). These variables were systematically excluded to maximize model precision. Fig 7 presents the VIF analysis results for all hazard-causing factors, illustrating that the final variable selection optimizes unique information content while minimizing statistical redundancy.

## 4.3 Hazard level assessment through comprehensive weighting

The inherent complexity of debris flow systems necessitates sophisticated factor weighting approaches that accommodate the multidimensional interactions among diverse variables. This study implements the game theory-based comprehensive weighting methodology detailed in section 2.1, achieving optimal integration of complementary weighting perspectives. The resulting factor weights derived from this approach are presented in Fig 8. The hazard level assessment procedure follows a systematic framework:

**Step 1:** Normalize the original dataset to eliminate the dimensional impact of different factors. The normalization formulas for maximum-type indicators and minimum-type indicators are shown in Equations (14) and (15), respectively. In Equations (14) – (15), $x_{ij}$ represents the $j^{th}$ influencing factor of the $i^{th}$ gully, $i = 1,2,…,n$ and $j = 1,2,…,m$, $n$ denotes the number of data groups, and $m$ denotes the number of factors.

$$r_{ij} = \frac{x_{ij} - min(x_{ij})}{max(x_{ij}) - min(x_{ij})} \tag{14}$$

$$r_{ij} = \frac{max(x_{ij}) - x_{ij}}{max(x_{ij}) - min(x_{ij})} \tag{15}$$

**Table 2. Hazard causation systems and their selection criteria.**

| Subsystem | Factor | Unit | Selection basis |
|---|---|---|---|
| Hydrometeorological subsystem | X1 | mm | Lee et al. [33] introduced an ANN model for predicting debris flows, identifying a nonlinear relationship between rainfall and the occurrence of debris flows. Higher annual average precipitation can create geotechnical conditions prone to debris flows, while high-intensity rainfall serves as a significant direct trigger for debris flow events. |
| | X2 | $10^3$mm | |
| | X3 | °C | Kidron et al. [34], Eppes et al. [35,36], and Deng et al. [37] highlighted the impacts of temperature, vapor pressure, humidity, and wind on rock weathering. During debris flows, the loosening of rocks due to weathering provides a substantial source of dynamic storage. Bai et al. [38,39] demonstrated that temperature and humidity affect the movement of soil particles. |
| | X4 | m/s | |
| | X5 | hPa | |
| | X6 | % | |
| Material source subsystem | X7 | $10^9$J/m² | Li [39] utilized this factor which integrates the impacts of elevation differences and dynamic source storage. Debris flows fundamentally involve transforming the gravitational potential energy of source material into kinetic energy, which quantifies the debris flow's destructive potential. |
| | X10 | – | Zhang et al. [40] employed the ratio of lengths in the sediment supply zone as a predictor for debris flow hazard levels. This metric indicates how unconsolidated materials are distributed within the riverbed, with homogeneous distribution increasing material availability. |
| | X19 | $10^4$/km² | Ma et al. [41] demonstrated that post-earthquake debris flow scale and frequency significantly increased due to augmented volumes of loose material sources caused by seismic activity. |
| Geographic subsystem | X8 | km² | Heiser et al. [42], Zhao et al. [43], and Li et al. [44] utilized catchment area as a key topographic factor. The catchment area determines the watershed's water-gathering capacity and sediment yield, making it a critical parameter universally considered in debris flow prediction models. |
| | X9 | – | Di et al. [45] incorporated the average gradient of the main channel in debris flow prediction. This parameter reflects channel slope, where a sufficient gradient is necessary for debris flow propagation, as low-slope conditions impede downward movement. |
| | X11 | km | Heiser et al. [42], Zhao et al. [43], and Li et al. [44] identified main channel length as a key parameter. Extended channels increase the potential for water/debris flows to transport larger quantities of loose material during descent. |
| | X12 | – | Kattel et al. [46] demonstrated through fluid dynamics analysis how meandering channels affect impact distance and solid material transport due to reduced flow velocity in curved sections. This coefficient quantifies channel sinuosity. |
| | X13 | Km/km² | Panchal et al. [47] established this factor's linear relationship with landslide probability using the Analytic Hierarchy Process (AHP) methodology. |
| | X14 | – | Shi et al. [48] applied contour roundness in debris flow analysis to characterize watershed geomorphology and hydrological properties. |
| | X15 | % | Zhao et al. [43] established through machine learning that vegetation cover significantly influences debris flow probability, with higher coverage substantially reducing occurrence likelihood. |
| | X16 | ° | Bertrand et al. [49] demonstrated that slope functions as a critical gravitational condition for debris flow formation while enhancing catchment water-collecting capacity. |
| | X17 | km | Heiser et al. [42], Zhao et al. [43], and Li et al. [44] utilized maximum relative elevation difference as a key parameter. This factor directly determines the gravitational potential energy available for conversion to kinetic energy, thus controlling the destructive capacity of resulting debris flows. |
| | X18 | – | Brardinoni et al. [50] established that lithology functions as a primary control mechanism for the generation rate of dynamic source materials in debris flow systems. |

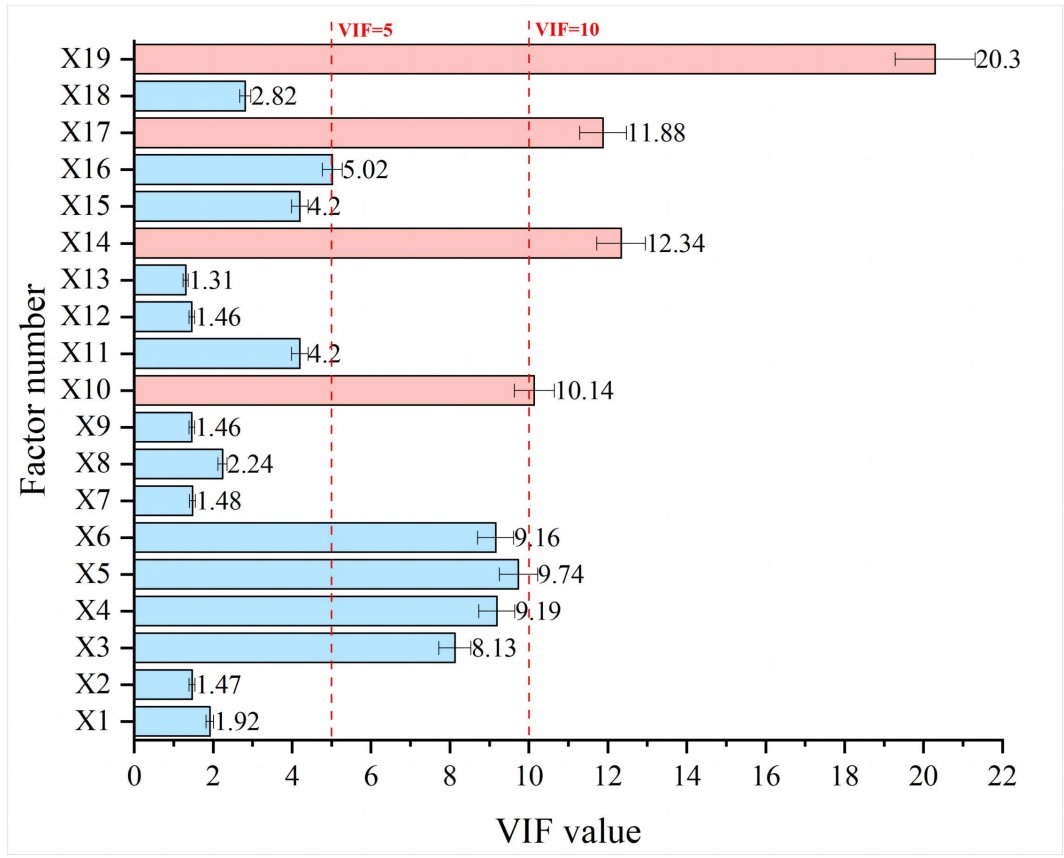

**Fig 7. VIF analysis results for hazard-causing factors.**

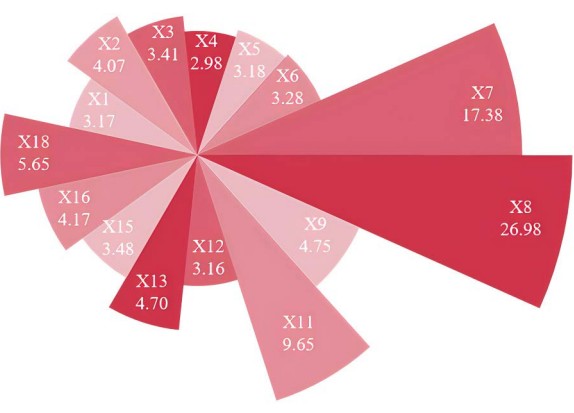

**Fig 8. Results of Comprehensive weighting.**

**Step 2:** For each debris flow gully, sum the weighted data of influencing factors to obtain its hazard assessment value. The calculation formula is shown in Equation (16). In Equation (16), wj represents the weight of each factor through Fig 8, and $r_{ij}$ denotes the normalized value of the $j^{th}$ influencing factor for the $i^{th}$ gully.

$$R_i = \sum_j r_{ij} \times \omega_j$$

<div align="right">(16)</div>

## 5 Results

To ensure statistical robustness, experimental parameters were standardized with population size fixed at 30 and iteration count at 100 across all optimization algorithms. The dataset underwent a 70%/30% training-testing partition, with results averaged across 50 independent runs. Notably, 12 test samples with distinctive geomorphological characteristics were deliberately included to evaluate model generalization capabilities across diverse debris flow conditions. Performance evaluation employed three complementary metrics: RMSE, MAE, and MAPE. The proposed IKOA-CNN-BiGRU-Attention model demonstrated exceptional predictive accuracy when benchmarked against 13 established machine learning regression models [52–55], as quantified in Table 3. Performance comparisons among the top five models are visualized in Fig 9. The hybrid model achieved unprecedented prediction precision with RMSE of $2.33 \times 10^{-6}$, MAE of $1.51 \times 10^{-6}$, and MAPE of 0.006%, representing multiple orders of magnitude improvement over conventional approaches.

This superior performance reflects systematic hyperparameter optimization through the improved Krill Herd algorithm. The learning rate underwent progressive refinement from 0.0061 to 0.0014, while GRU units converged to 116, CNN filters stabilized at 54, and batch size optimized at 31. The hyperparameter evolution across iterations, detailed in Table 4, demonstrates the algorithm's capacity to navigate complex parameter spaces while avoiding local optima. Architecturally, the bidirectional GRU component captures temporal dependencies in multidimensional feature sequences, while the attention mechanism adaptively prioritizes critical factors including slope characteristics, precipitation parameters, and drainage properties. Unlike models such as GWO-ELM and PSO-SVM, which exhibit performance degradation with geomorphologically distinct test cases, the IKOA-CNN-BiGRU-Attention model maintains exceptional prediction stability across diverse geological contexts, validating its effectiveness in modeling the inherently nonlinear and multidimensional nature of debris flow hazard systems.

## 6 Results explanation and disaster mitigation implications

### 6.1 Feature importance analysis and threshold identification in debris flow genesis

Through systematic application of SHAP methodology, this study deconstructs the internal decision processes of the IKOA-CNN-BiGRU-Attention model, revealing that the architecture captures both physical threshold effects and

**Table 3. The prediction error of each model.**

| No. | Model | RMSE | MAE | MAPE |
|---|---|---|---|---|
| 1 | IKOA-CNN-BiGRU-Attention | 2.33E-06 | 1.51E-06 | 0.006% |
| 2 | PSO-SVM | 0.0691 | 0.0352 | 26.74% |
| 3 | GWO-ELM | 0.0017 | 0.0008 | 1.98% |
| 4 | IDBO-VMD-BiLSTM | 0.0636 | 0.0451 | 60.57% |
| 5 | DT | 0.1024 | 0.0522 | 43.24% |
| 6 | Adaboost | 0.1030 | 0.0520 | 117.84% |
| 7 | IAO-BiLSTM | 0.0582 | 0.0333 | 73.20% |
| 8 | RF | 0.1155 | 0.0610 | 78.35% |
| 9 | GA-BP | 0.3155 | 0.2359 | 17.01% |
| 10 | GBDT | 0.1060 | 0.0540 | 133.58% |
| 11 | Extra trees | 0.1060 | 0.0540 | 132.10% |
| 12 | CatBoost | 0.1170 | 0.0630 | 319.26% |
| 13 | KNN | 0.1152 | 0.0630 | 298.79% |
| 14 | XGBoost | 0.0990 | 0.0500 | 99.22% |

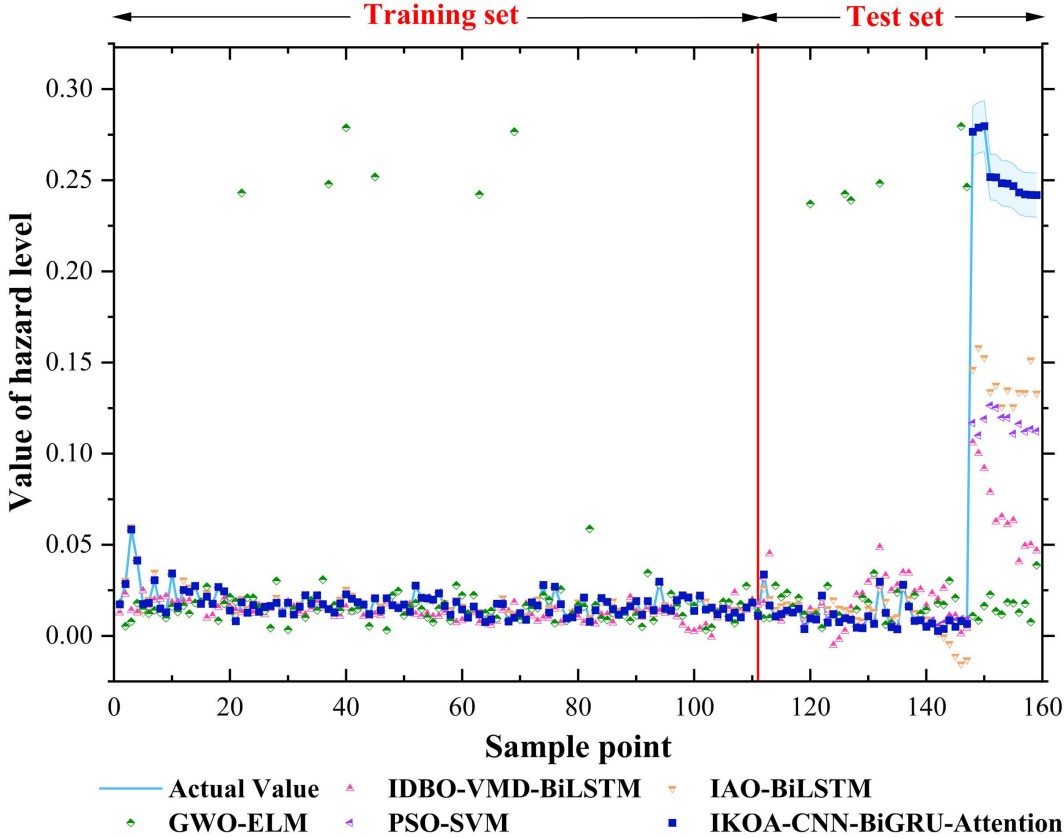

**Fig 9. Predictive performance of the training and test sets.**

**Table 4. Results for model hyperparameter values.**

| Iteration | Learning Rate | GRU Units | CNN Filters | Batch Size | Loss |
|---|---|---|---|---|---|
| 5 | 0.0061 | 90 | 42 | 26 | 0.2501 |
| 15 | 0.0048 | 105 | 48 | 28 | 0.2157 |
| 28 | 0.0032 | 112 | 50 | 30 | 0.1954 |
| 40 | 0.0024 | 116 | 52 | 31 | 0.1878 |
| 60 | 0.0019 | 120 | 54 | 32 | 0.1843 |
| 80 | 0.0014 | 116 | 54 | 31 | 0.1825 |
| 95 | 0.0014 | 116 | 54 | 31 | 0.1825 |
| Optimization range | [0.0001, 0.01] | [32, 128] | [16, 64] | [16, 64] | – |

multifactorial regulatory relationships governing debris flow initiation. Fig 10(a) presents a quantitative hierarchical ranking of feature importance wherein potential source energy X7 emerges as the predominant factor with a mean absolute SHAP value of $1.07 \times 10^{-2}$, approximately 4.6-fold greater than the second-ranked parameter. Annual mean temperature X3, annual mean vapor pressure X5, and average slope X16 constitute the second tier of importance, while annual mean rainfall X2, vegetation coverage X15, and main channel length X11 form a distinct third tier.

Fig 10(b) elucidates the dependency relationship between potential source energy X7 and its corresponding SHAP values, demonstrating a threshold effect in debris flow triggering mechanisms. The relationship manifests as a sigmoidal

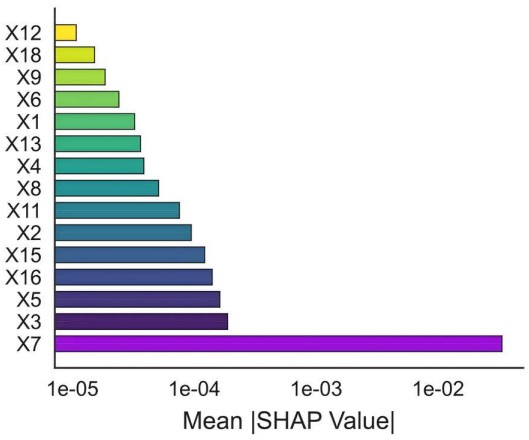

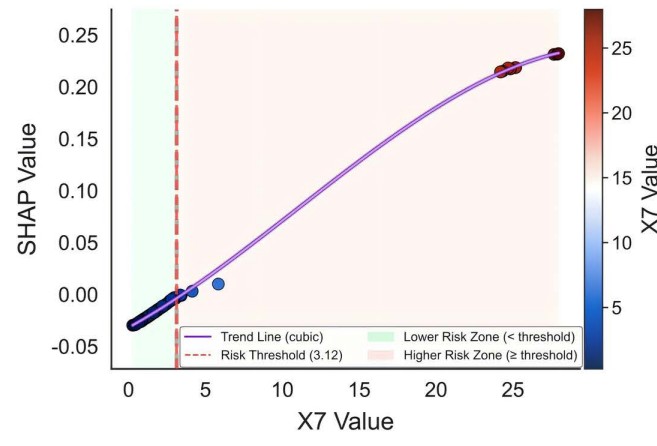

(a) Feature importance ranking with logarithmic scale    (b) SHAP dependence plot for X7 feature

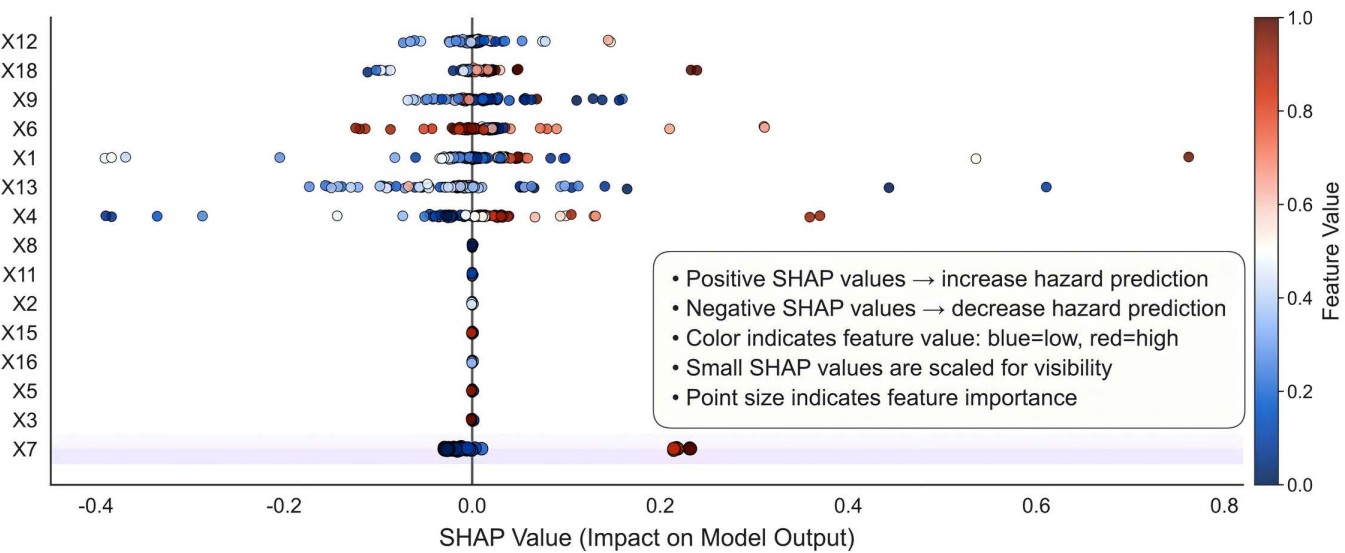

(c) Comprehensive SHAP value distribution across all features with magnitude impact

**Fig 10. SHAP analysis of feature importance and threshold effects in debris flow formation.**

nonlinear curve, optimally fitted by the cubic function $y = 1.25 \times 10^{-4}x^3 - 9.37 \times 10^{-4}x^2 + 1.83 \times 10^{-2}x - 2.21 \times 10^{-2}$ with an $R^2$ value of 0.98. A critical inflection point is identified at $X7 \approx 3.12 \times 10^9$ J/m², effectively partitioning the risk domain into two distinct regions: a low-risk zone where $X7 < 3.12 \times 10^9$ J/m² and a high-risk zone where $X7 > 3.12 \times 10^9$ J/m². Within the low-risk zone, SHAP values consistently approximate zero or exhibit slightly negative values, indicating insufficient energy accumulation for debris flow mobilization; within the high-risk zone, SHAP values demonstrate an accelerating positive trend as X7 increases, reflecting the nonlinear amplification characteristic of debris flow initiation.

The SHAP value distribution in Fig 10(c) further illuminates the distinct influence patterns of various factors on model predictions. The distribution for X7 exhibits pronounced bimodal segregation: low-value samples cluster tightly around the zero SHAP value region, while high-value samples display substantial positive shifts, reaching SHAP values exceeding 0.2, surpassing the influence range of all other features. This polarized distribution provides statistical confirmation of

the threshold phenomenon. High values of maximum 24-hour rainfall X1 and annual mean relative humidity X6 similarly exhibit positive SHAP value tendencies, albeit with considerably greater dispersion, indicating that these parameters function as conditional triggers whose efficacy is contingent upon the antecedent state of source energy accumulation.

Based on the comprehensive SHAP analysis, a physically explainable mechanistic framework for debris flow formation emerges: potential source energy X7 constitutes the fundamental prerequisite condition with a quantified threshold of $3.12 \times 10^9$ J/m$^2$; environmental parameters such as annual mean temperature X3 and annual mean vapor pressure X5 function as regulatory modulators, exerting indirect effects by altering source material stability; hydrological factors, primarily maximum 24-hour rainfall X1, operate as activation triggers that initiate debris flow mobilization only when the prerequisite energy threshold condition has been satisfied.

## 6.2 Factor interaction dynamics and risk response characterization

Building upon the identification of potential source energy X7 as the dominant factor, this section systematically explores factor interactions and their impact on debris flow risk prediction, revealing complex relationships with implications for monitoring and early warning systems optimization. Fig 11 illustrates the interaction effects between source energy and key environmental variables. The interaction between X7 and annual mean temperature X3 reveals a significant correlation $r = 0.06$, $p < 0.05$, with risk contribution substantially increasing when the temperature exceeds 16°C in high source energy areas. This temperature-energy interaction suggests that thermal conditions may accelerate material destabilization processes in areas with significant loose material accumulation. The practical implication for disaster prevention is that temperature monitoring should be integrated into early warning systems, particularly for regions with high material accumulation. Warning thresholds may require seasonal adjustments with heightened vigilance during warmer periods when temperatures exceed 16°C. Similarly, the interaction between X7 and annual mean vapor pressure X5 demonstrates a stronger correlation $r = 0.19$, $p < 0.01$, with high X7 samples manifesting significantly elevated SHAP values when X5 exceeds 15 hPa. This finding indicates that humidity sensors should be incorporated into monitoring networks alongside traditional rainfall gauges, particularly in areas with substantial loose material accumulation, as atmospheric moisture conditions appear to play an important role in destabilizing source materials.

The interaction between maximum 24-hour rainfall X1 and X7 provides particularly valuable insights for risk management. Despite statistical non-significance in overall correlation $r = 0.02$, $p > 0.05$, clear nonlinear patterns emerge: rainfall intensity substantially impacts risk only when source energy exceeds approximately $5 \times 10^9$ J/m$^2$, while having minimal effect below this threshold. This conditional relationship fundamentally challenges the conventional rainfall-threshold approach to debris flow warning. For early warning system design, this finding suggests implementing a hierarchical approach where areas are first classified by source energy potential, with advanced rainfall monitoring prioritized specifically in high-energy zones. For low-energy zones, basic rainfall monitoring may be sufficient regardless of precipitation intensity. Rainfall thresholds should be dynamic rather than fixed, adjusting based on source material conditions.

Fig 12 refines the threshold identification through a detailed SHAP response analysis. The upper panel presents a higher-order polynomial analysis of the X7 SHAP response curve with $R^2 = 0.99$, identifying a critical inflection point at $X7 = 3.00 \times 10^9$ J/m$^2$. This value provides a quantitative criterion for classifying terrain vulnerability in hazard mapping exercises and prioritizing areas for source material control measures such as check dams, retaining walls, and slope stabilization projects. The upper right panel identifies a distinct inflection point in the X1 SHAP dependency curve at $X1 = 27.01$ mm, providing a data-driven basis for rainfall warning standards. This specific threshold offers a quantitative reference for evaluating and potentially adjusting current warning thresholds in operational early warning systems. The lower panel integrates these insights into a comprehensive two-dimensional risk contour map with three distinct risk domains that translate directly to operational warning levels. The blue region represents normal conditions where predicted risk values fall below 0.03, warranting routine monitoring procedures. The yellow zone indicates heightened alert conditions with

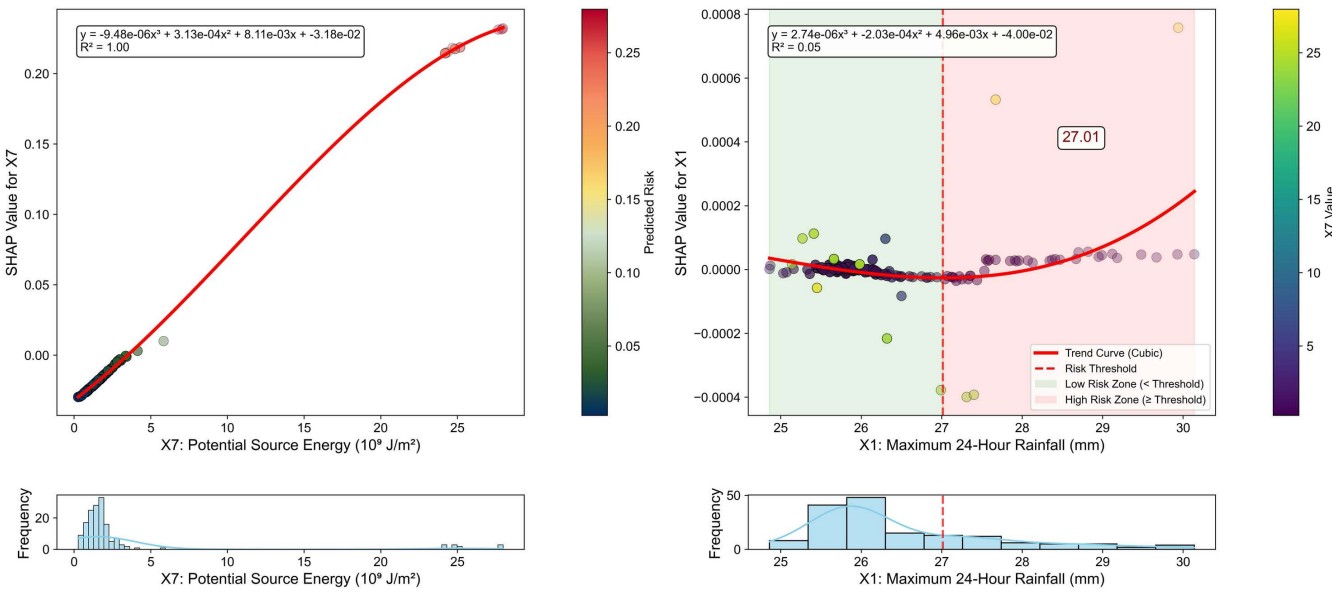

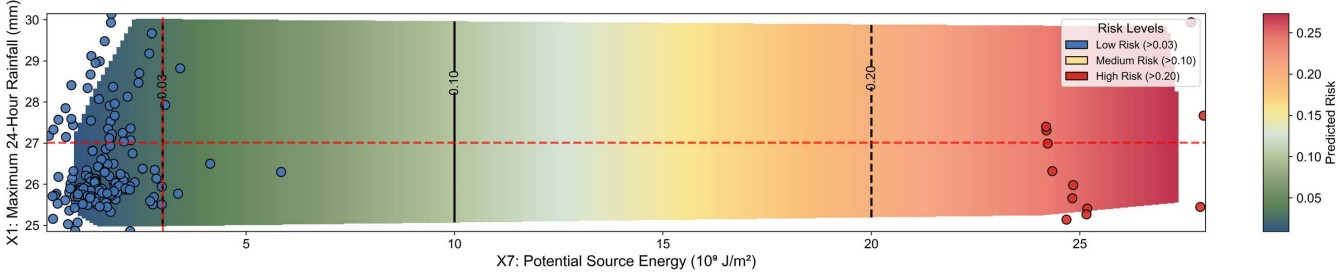

**Fig 11. SHAP interactions between key geomorphological and meteorological variables.**

predicted risk values between 0.03 and 0.20, necessitating increased monitoring frequency and preliminary emergency preparations. The red domain signifies emergency response conditions where the predicted risk exceeds 0.20, requiring immediate protective actions including potential evacuations.

Fig 13 extends this analysis into three dimensions, revealing a comprehensive risk response surface that captures the compound effects of X7 and X1. The visualization of threshold planes creates four distinct quadrants that translate to operational intervention priority zones. The first quadrant, characterized by values below both X7 and X1 thresholds, indicates low-priority zones warranting only basic monitoring. The second quadrant, where X7 exceeds its threshold while X1 remains below its critical value, demarcates source-control priority zones where engineering measures to reduce material accumulation should be emphasized. The third quadrant, characterized by X1 values above the threshold while X7 remains below its critical level, identifies hydrological management priority zones where drainage improvements and runoff control should be prioritized. The fourth quadrant, where both X7 and X1 exceed their respective thresholds, delineates comprehensive intervention zones requiring integrated approaches combining multiple strategies.

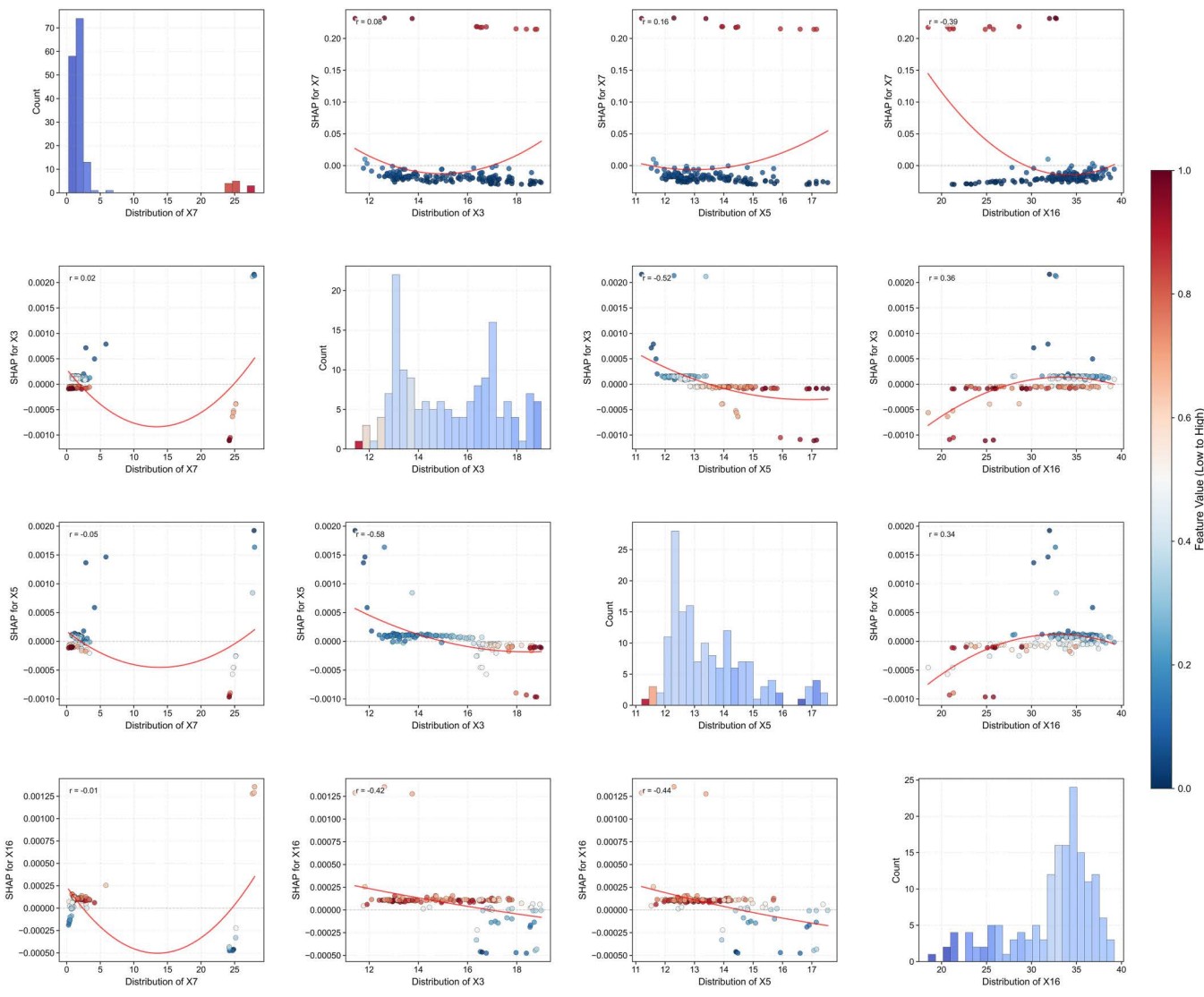

**Fig 12. Threshold identification for debris flows through SHAP response analysis.**

## 6.3 Intervention efficacy assessment and risk reduction strategy optimization

Building on identified risk factors and thresholds, this section evaluates the comparative efficacy of different risk reduction approaches, providing quantitative insights for evidence-based disaster management decision-making. Figs 14(a) and 14(b) present comparative risk response curves for interventions targeting X7 and X1, respectively. The analysis reveals that reducing maximum 24-hour rainfall X1 from 29.5 mm to 27.0 mm yields a 58.8% risk reduction, while reducing potential source energy X7 from $8.0 \times 10^9$ J/m² to $3.0 \times 10^9$ J/m² achieves a 25.0% risk diminution. This substantial difference in intervention efficacy provides clear guidance for resource allocation: rainfall control measures may yield faster risk reduction returns compared to source control measures. Figs 14(c) and 14(d) further refine this analysis through spatially-resolved risk reduction percentage contour maps, revealing that vulnerability mapping should identify transition zones where risk factors are marginally above threshold values, as modest interventions in these areas may yield substantial risk reduction.

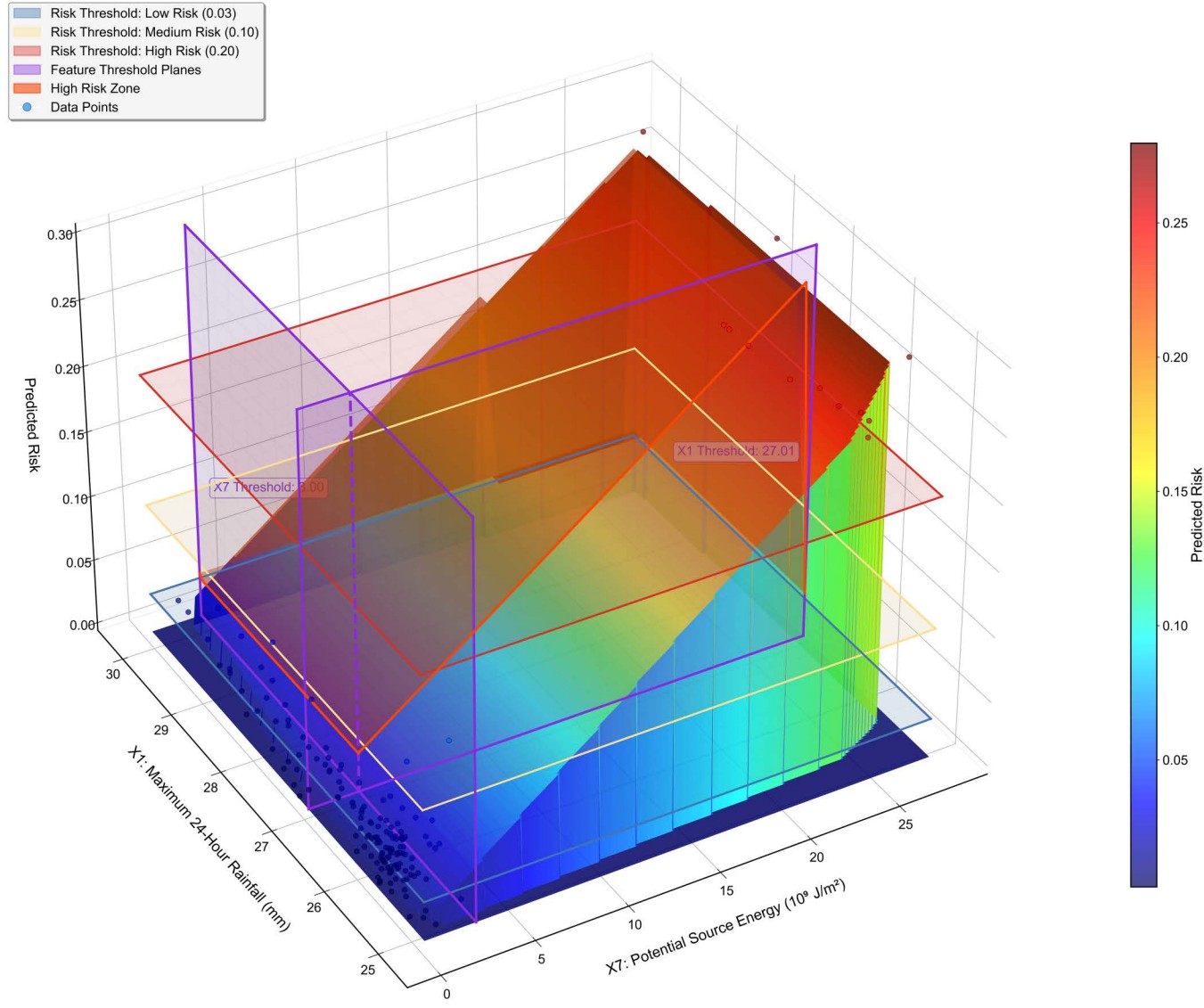

**Fig 13. Three-dimensional risk surface with critical threshold planes.**

Fig 15 presents SHAP force plots from representative data points selected through stratified random sampling, revealing substantial heterogeneity in localized risk factor compositions. This heterogeneity indicates that effective risk management requires location-specific approaches tailored to the dominant local risk factors. For implementation, this suggests customizing monitoring systems based on local risk factor compositions. Locations dominated by meteorological factors would benefit from enhanced weather-based warning systems with meteorological monitoring, while areas where topographical factors predominate would likely respond better to physical infrastructure interventions rather than precipitation-based warning systems.

Synthesizing these analytical results, a practical framework for precision debris flow risk management emerges with several actionable components. Early warning systems should implement rainfall thresholds that adjust based on source energy conditions rather than universal rainfall values, with lower rainfall thresholds applied in high source energy areas

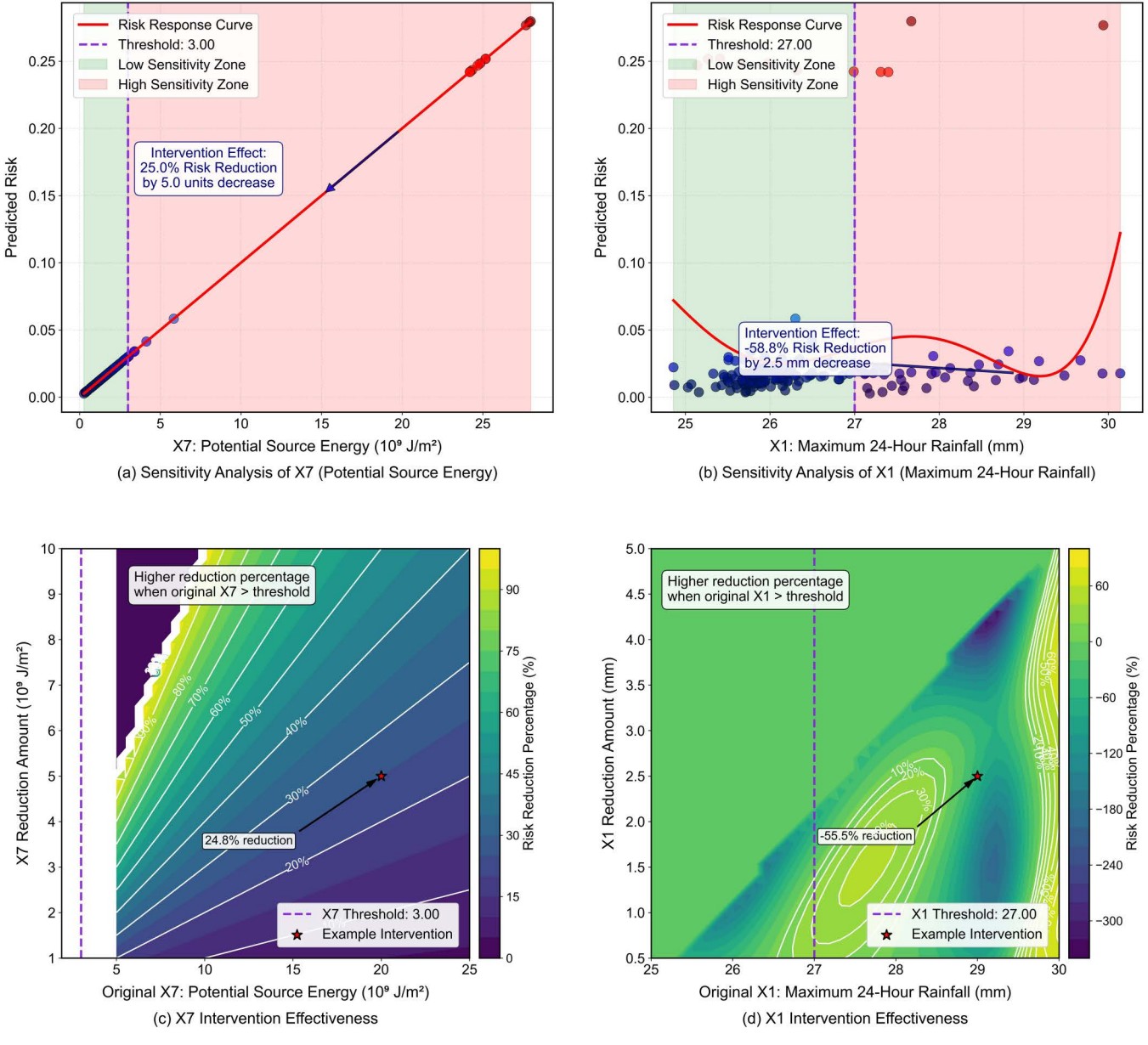

**Fig 14. Comparative efficiency of source energy versus rainfall control interventions.**

and higher thresholds in low energy areas. Beyond traditional rainfall monitoring, temperature and humidity sensors should be incorporated into monitoring networks, especially in areas where source energy exceeds the identified threshold. In resource-constrained environments, interventions should be sequenced according to threshold proximity, prioritizing areas just above threshold values where modest investments yield substantial risk reduction. Different zones require specific approaches: high-source energy zones warrant engineering interventions to reduce source material accumulation; critical rainfall zones require drainage improvements and runoff control; dual-threshold exceedance zones necessitate comprehensive management strategies incorporating both engineering measures and enhanced monitoring systems. Warning thresholds and monitoring intensity should be adjusted seasonally where temperature appears as a significant

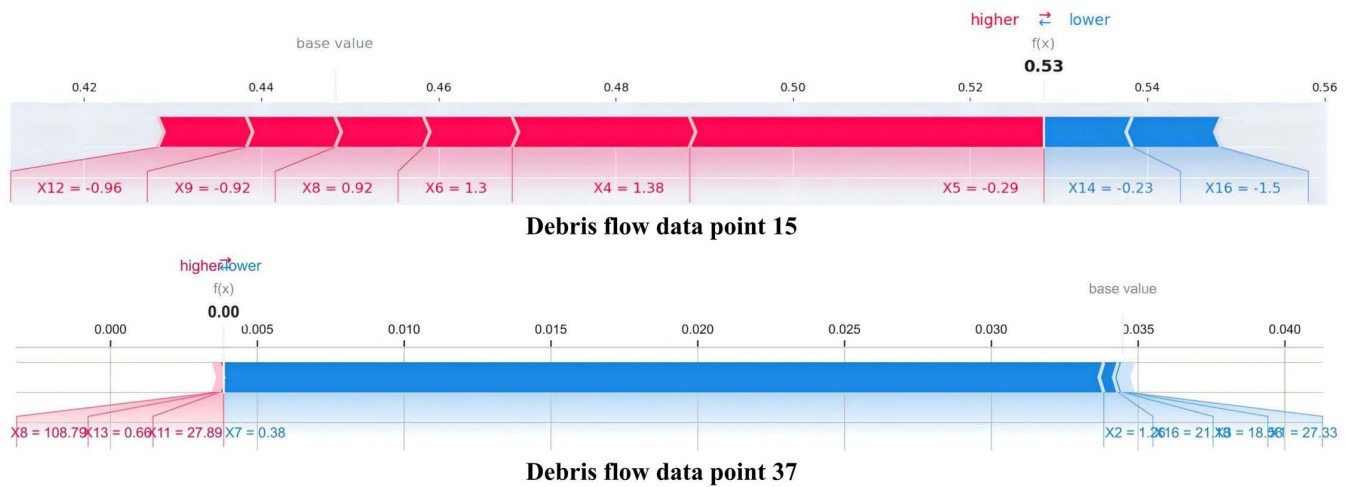

**Fig 15. Site-specific risk factor compositions in contrasting debris flow scenarios.**

risk modifier, with heightened vigilance during periods exceeding 16°C. While the specific threshold values identified in this study are calibrated to the study area's geological and climatological context, the methodological framework for threshold identification and factor interaction analysis remains transferable across diverse geographical settings, allowing regional calibration of specific thresholds while maintaining the structural integrity of the factor interaction paradigm.

## 7  Discussions

### 7.1.  Stability analysis

The stability of a model is crucial for reliable debris flow hazard prediction, as it ensures consistent results across multiple independent runs. To evaluate the stability of the IKOA-CNN-BiGRU-Attention model, 50 independent experiments were conducted, recording the Root Mean Square Error (RMSE), Mean Absolute Error (MAE), and their standard deviations (SD). Lower SD values indicate higher stability and consistency in predictions. As shown in Table 5 the IKOA-CNN-BiGRU-Attention model demonstrated minimal performance fluctuations, with an RMSE of $2.33 \times 10^{-6} \pm 1.51 \times 10^{-7}$ and an MAE of $1.51 \times 10^{-6} \pm 1.10 \times 10^{-7}$, outperforming all comparative models. These results highlight the model's reliability and reproducibility, making it highly suitable for practical applications in debris flow hazard prediction. Its stable performance improves the credibility of risk assessments and provides robust technical support for disaster prevention and mitigation.

### 7.2.  Robustness analysis

Robustness is essential for ensuring a model's reliability when handling data disturbances such as noise or outliers, which are common in real-world geological and meteorological datasets. To evaluate the robustness of the IKOA-CNN-BiGRU-Attention model, random noise levels of 5%, 10%, and 20% were added to the test dataset to simulate possible data disturbances. The changes in performance metrics, including Root Mean Square Error (RMSE) and Mean Absolute Error (MAE), were observed to determine the model's ability to handle noisy data. As shown in Table 6, the IKOA-CNN-BiGRU-Attention model showed minimal performance degradation. When the noise level increased from 0% to 20%, the RMSE rose by only $7.9 \times 10^{-7}$, and the MAE increased by approximately $4.9 \times 10^{-7}$. Compared to other models, these slight increases demonstrate the strong robustness of the proposed model, which ensures high predictive accuracy even under noisy data conditions. This robustness confirms the model's reliability in complex environments and enhances its applicability to practical disaster prevention and mitigation efforts.

**Table 5. Stability analysis of each model.**

| No. | Model | RMSE±SD | MAE±SD |
|---|---|---|---|
| 1 | IKOA-CNN-BiGRU-Attention | 2.33E-06±1.51E-07 | 1.51E-06±1.10E-07 |
| 2 | PSO-SVM | 0.0691±0.0020 | 0.0352±0.0015 |
| 3 | GWO-ELM | 0.0017±0.00015 | 0.0008638±0.00012 |
| 4 | IDBO-VMD-BiLSTM | 0.0636±0.0012 | 0.0451±0.0011 |
| 5 | DT | 0.1024±0.0030 | 0.0522±0.0023 |
| 6 | Adaboost | 0.103±0.0035 | 0.052±0.0028 |
| 7 | IAO-BiLSTM | 0.0582±0.0015 | 0.0333±0.0013 |
| 8 | RF | 0.1155±0.004 | 0.0610±0.0032 |
| 9 | GA-BP | 0.3155±0.007 | 0.2359±0.0055 |
| 10 | GBDT | 0.106±0.0038 | 0.054±0.0030 |
| 11 | Extra trees | 0.106±0.0039 | 0.054±0.0031 |
| 12 | CatBoost | 0.117±0.0041 | 0.063±0.0034 |
| 13 | KNN | 0.115±0.0040 | 0.063±0.0034 |
| 14 | XGBoost | 0.099±0.0028 | 0.050±0.0025 |

**Table 6. Robustness Analysis of each model.**

| Model | 5%Noise | | 10%Noise | | 20%Noise | |
|---|---|---|---|---|---|---|
| | RMSE | MAE | RMSE | MAE | RMSE | MAE |
| IKOA-CNN-BiGRU-Attention | 2.60E-06 | 1.65E-06 | 2.95E-06 | 1.80E-06 | 3.12E-06 | 2.00E-06 |
| PSO-SVM | 0.0720 | 0.0370 | 0.0780 | 0.0410 | 0.0850 | 0.0450 |
| GWO-ELM | 0.0018 | 0.0008 | 0.0019 | 0.0009 | 0.0021 | 0.0011 |
| IDBO-VMD-BiLSTM | 0.0650 | 0.0470 | 0.0720 | 0.0510 | 0.0810 | 0.0570 |
| DT | 0.0610 | 0.0370 | 0.0680 | 0.0420 | 0.0760 | 0.0490 |
| Adaboost | 0.1100 | 0.0560 | 0.1250 | 0.0630 | 0.1440 | 0.0740 |
| IAO-BiLSTM | 0.1080 | 0.0540 | 0.1200 | 0.0610 | 0.1320 | 0.0690 |
| RF | 0.1220 | 0.0640 | 0.1380 | 0.0710 | 0.1550 | 0.0820 |
| GA-BP | 0.3250 | 0.2420 | 0.3600 | 0.270 | 0.4100 | 0.3150 |
| GBDT | 0.1120 | 0.0570 | 0.1250 | 0.0630 | 0.1400 | 0.0710 |
| Extra trees | 0.1150 | 0.0590 | 0.1300 | 0.0650 | 0.1470 | 0.0750 |
| CatBoost | 0.1280 | 0.0670 | 0.1400 | 0.0750 | 0.1600 | 0.0840 |
| KNN | 0.1250 | 0.0650 | 0.1380 | 0.0730 | 0.1550 | 0.0830 |
| XGBoost | 0.1050 | 0.0540 | 0.1180 | 0.0600 | 0.1350 | 0.0700 |

## 7.3. Applicability analysis

The applicability of a model is crucial for assessing its practical value, particularly its suitability across different regions and datasets. To evaluate the applicability of the IKOA-CNN-BiGRU-Attention model, a 5-fold cross-validation experiment was conducted. The Root Mean Square Error (RMSE) and Mean Absolute Error (MAE) were recorded for each fold to assess performance consistency. As shown in the results in Table 7, the RMSE ranged from $2.35 \times 10^{-6}$ to $2.50 \times 10^{-6}$, and the MAE ranged from $1.55 \times 10^{-6}$ to $1.65 \times 10^{-6}$, with minimal fluctuations across the folds. This consistent performance demonstrates the model's strong applicability and stability, making it suitable for predicting debris flow hazards not only in the specific study area but also in other geographical regions. Its generalizability provides robust technical support for global disaster risk assessment and early warning efforts.

**Table 7. Applicability analysis of the IKOA-CNN-BiGRU-Attention model.**

| Number of cross-validations | RMSE | MAE |
|---|---|---|
| 1st Fold | 2.40E-06 | 1.60E-06 |
| 2nd Fold | 2.50E-06 | 1.65E-06 |
| 3rd Fold | 2.35E-06 | 1.55E-06 |
| 4th Fold | 2.42E-06 | 1.62E-06 |
| 5th Fold | 2.45E-06 | 1.63E-06 |

## 8 Conclusions

This study proposes an integrated IKOA-CNN-BiGRU-Attention framework with SHAP explainability for high-precision debris flow hazard prediction that combines advanced optimization techniques, multidimensional deep learning architectures, and explainable AI mechanisms. Through systematic implementation and validation in the Nujiang River Basin, China, the study yields five principal contributions:

(1) The Improved Kepler Optimization Algorithm was developed with three key innovations: Chebyshev mapping for ergodic population initialization, golden sine operator for solution space narrowing, and dynamic weight coefficient for balanced search dynamics. Benchmark function testing demonstrated IKOA's superior performance across standard optimization cases, consistently outperforming conventional algorithms including GWO, SSA, and WOA. This optimization algorithm, when integrated with the CNN-BiGRU-Attention architecture, achieved exceptional prediction performance with an RMSE of $2.33 \times 10^{-6}$, MAE of $1.51 \times 10^{-6}$, and MAPE of 0.006%, outperforming all 13 benchmark models.

(2) Comprehensive validation confirmed the framework's reliability and generalizability for real-world implementation. Stability analysis through 50 independent experimental runs demonstrated consistent performance with RMSE of $2.33 \times 10^{-6} \pm 1.51 \times 10^{-7}$, MAE of $1.51 \times 10^{-6} \pm 1.10 \times 10^{-7}$. Robustness assessment under noise perturbation showed minimal performance degradation with RMSE increasing by only $7.9 \times 10^{-7}$ at 20% noise injection, confirming the model's resilience to data uncertainty. Five-fold cross-validation further validated generalizability with RMSE ranging from $2.35 \times 10^{-6}$ to $2.50 \times 10^{-6}$, suggesting good applicability across diverse geomorphological contexts.

(3) SHAP explainability analysis revealed a previously unrecognized hierarchical triggering mechanism in debris flow genesis. Potential source energy emerged as the primary factor with a mean absolute SHAP value of $1.07 \times 10^{-2}$, approximately 4.6-fold greater than other parameters. The relationship between source energy and debris flow hazard exhibits a sigmoidal pattern fitted by a cubic function ($R^2 = 0.98$) with a critical threshold at $3.00 \times 10^9$ J/m$^2$. This threshold partitions the risk domain into two distinct regions: a low-risk zone where hazard potential remains minimal regardless of other factors, and a high-risk zone where hazard potential increases nonlinearly as source energy rises.

(4) The analysis identified a "dual-threshold conditional triggering" framework for debris flow hazard prediction. The maximum 24-hour rainfall threshold was established at 27.01 mm; however, this threshold's effectiveness appears to depend on source energy conditions. When source energy remains below $5 \times 10^9$ J/m$^2$, rainfall intensity shows minimal impact on hazard potential; once this energy threshold is exceeded, rainfall appears to function as an activation trigger.

(5) Intervention efficacy assessment quantified the differential impact of alternative risk reduction strategies. Rainfall control measures produced a 58.8% risk reduction when decreasing maximum 24-hour rainfall from 29.5 mm to 27.0 mm, compared to source energy interventions which achieved a 25.0% risk reduction when decreasing potential source energy from $8.0 \times 10^9$ J/m$^2$ to $3.0 \times 10^9$ J/m$^2$. These findings provide evidence-based guidance for resource allocation

in disaster risk reduction programs, particularly for identifying transition zones where modest interventions may yield substantial risk reduction.

## Acknowledgments

The authors would like to acknowledge the support provided by the Information Technology Center, Zhejiang University.

## Author contributions

**Conceptualization:** Tianlong Wang.

**Data curation:** Tianlong Wang.

**Formal analysis:** Hao Yang, Tianlong Wang.

**Funding acquisition:** Tianlong Wang, Nikita Igorevich Fomin.

**Investigation:** Tianlong Wang, Nikita Igorevich Fomin, Shuoting Xiao, Liang Liu.

**Methodology:** Hao Yang.

**Project administration:** Tianlong Wang, Shuoting Xiao, Liang Liu.

**Resources:** Tianlong Wang.

**Software:** Hao Yang.

**Supervision:** Tianlong Wang, Nikita Igorevich Fomin, Shuoting Xiao.

**Validation:** Hao Yang.

**Visualization:** Hao Yang, Tianlong Wang, Nikita Igorevich Fomin, Shuoting Xiao, Liang Liu.

**Writing – original draft:** Hao Yang, Tianlong Wang.

**Writing – review & editing:** Hao Yang, Tianlong Wang, Nikita Igorevich Fomin, Shuoting Xiao, Liang Liu.

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
