## [Decision Letter · Decision Letter 0]

Dear Dr. Wang,

Thank you for submitting your manuscript to PLOS ONE. After careful consideration, we feel that it has merit but does not fully meet PLOS ONE’s publication criteria as it currently stands. Therefore, we invite you to submit a revised version of the manuscript that addresses the points raised during the review process.

The review process is now complete, and we have two reports submitted by the expert reviewers. As can be seen from these reports, one of them requested revision before the manuscript was accepted. However, another explicitly recommended rejection without revision. The author must be aware that the comments of reviewers #2 are highly critical of the overall study. Therefore, I suggest the authors carefully revise this manuscript point by point according to all the comments. The resubmitted manuscript will be reviewed again.

In the end, I want to thank the reviewers for their great efforts on your manuscript and you for your submission.

We look forward to receiving your revised manuscript.

Kind regards,

Linwei Li

Academic Editor

PLOS ONE

4. We note that Figure 6 in your submission contain [map/satellite] images which may be copyrighted. All PLOS content is published under the Creative Commons Attribution License (CC BY 4.0), which means that the manuscript, images, and Supporting Information files will be freely available online, and any third party is permitted to access, download, copy, distribute, and use these materials in any way, even commercially, with proper attribution. For these reasons, we cannot publish previously copyrighted maps or satellite images created using proprietary data, such as Google software (Google Maps, Street View, and Earth). For more information, see our copyright guidelines: http://journals.plos.org/plosone/s/licenses-and-copyright.

1. You may seek permission from the original copyright holder of Figure 6 to publish the content specifically under the CC BY 4.0 license. 

Additional Editor Comments:

None

Reviewers' comments:

Reviewer's Responses to Questions

**Comments to the Author**

1. Is the manuscript technically sound, and do the data support the conclusions?

Reviewer #1: Yes

Reviewer #2: No

2. Has the statistical analysis been performed appropriately and rigorously?

Reviewer #1: Yes

Reviewer #2: No

3. Have the authors made all data underlying the findings in their manuscript fully available?

Reviewer #1: Yes

Reviewer #2: No

4. Is the manuscript presented in an intelligible fashion and written in standard English?

Reviewer #1: Yes

Reviewer #2: Yes

Reviewer #1: This manuscript presents an integrated framework that combines IKOA-CNN-BiGRU-Attention architecture with SHAP explainability analysis for debris flow hazard prediction in the Nujiang River Basin, China. The research demonstrates innovation in the field of debris flow hazard prediction, particularly in the integration of optimization algorithms, deep learning architectures, and explainability mechanisms. However, some methodological issues should be addressed, particularly regarding the validation of prediction accuracy, reconciliation of variable importance rankings, and justification of analytical choices. I recommend revision before the manuscript can be considered for publication:

(1) The introduction's exposition of five paradigms in debris flow prediction research requires structural refinement. The transition between the discussion of the "black box problem" (lines 87-94) and XAI methodologies (lines 95-103) lacks sufficient logical connectivity, thereby undermining the theoretical justification for adopting SHAP methodology in this investigation. I recommend incorporating appropriate transitional discourse to articulate explicitly why SHAP methodology constitutes an optimal approach for addressing the research questions, and to establish a more robust conceptual linkage between the chosen methodology and the research objectives.

(2) The experimental results presented in Table 3 indicate that the IKOA-CNN-BiGRU-Attention model achieves remarkably low error metrics: RMSE of 2.33×10-6, MAE of 1.51×10-6, and MAPE of 0.006%. Given the inherent complexity and stochasticity of debris flow phenomena, characterized by non-linear processes and substantial data uncertainties, such extraordinarily precise predictions warrant careful scrutiny. The authors should provide a comprehensive explanation of the validation procedures employed to ensure result reliability, discuss potential limitations of the model when deployed in real-world scenarios, and elucidate the practical significance of such high-precision forecasting capabilities for operational debris flow warning systems.

(3) A salient discrepancy exists between the SHAP global explanation analysis (lines 449-479) and the comprehensive weighting results (lines 408-425). Figure 8 identifies potential source energy (X7, 26.98%), catchment area (X8, 17.38%), and main channel length (X11, 9.65%) as the predominant influential variables, whereas Figure 10's SHAP analysis presents a divergent hierarchical ordering, with X1 (maximum 24-hour rainfall) and X2 (annual average rainfall) occupying the second and third positions, respectively. This inconsistency in factor importance ranking necessitates a thorough explication to reconcile these seemingly contradictory findings and to strengthen the credibility of the research conclusions.

(4) The SHAP local explanation analysis (lines 480-507) selectively examines only sample points 15 and 37, without providing any methodological rationale for this selection. If these particular sample points were chosen as representative case studies, their geological characteristics and typological representativeness should be explicitly delineated; alternatively, if they were randomly selected, the sampling methodology should be transparently documented to mitigate concerns regarding selection bias. Enhancement of this section will substantially augment the persuasiveness of the SHAP analytical findings.

(5) The multicollinearity diagnostic procedure (lines 385-394) employs VIF<5 as a threshold criterion, rather than the more conventionally accepted standard of VIF<10 in statistical literature. The authors should provide a theoretical or empirical justification for adopting this more stringent threshold, and should include the original VIF values for the eliminated variables (X10, X14, X17, and X19) to enable readers to independently evaluate the validity of the variable selection process. Such transparency will enhance the methodological rigor and reproducibility of the research.

(6) The bibliographic apparatus exhibits significant inconsistencies in formatting protocol. References 2, 3, and 4 lack requisite Digital Object Identifier information, whereas subsequent entries incorporate complete digital identifiers. It is imperative that the authors systematically incorporate standardized DOI information for all references in accordance with established journal citation guidelines. Such standardization not only conforms to contemporary bibliographic conventions but also facilitates efficient access to primary literature sources for the scholarly audience.

(7) Citation methodologies within the manuscript text demonstrate concerning inconsistency in application. Line 44 employs the author-date format "Berti and Simoni (2007)," while alternative sections utilize numerical references enclosed in square brackets. Scholarly writing necessitates methodological consistency; therefore, the authors must adopt a uniform citation system throughout the manuscript. This standardization should adhere to established academic protocols, which demand consistency in referential apparatus as a fundamental aspect of scholarly communication.

(8) Reference citation practices delineated in Table 2 manifest substantial heterogeneity in presentational format. Certain entries present full author nomenclature such as "Deuk-Hwan Lee," while others utilize surnames accompanied by temporal indicators such as "Bing Bai in 2020." This inconsistency contravenes established bibliographic principles. Standardization of citation format in accordance with the target journal's stylistic requirements is essential to maintain scholarly integrity and enhance the academic presentation of the manuscript.

(9) Table 2 incorporates the abbreviated designation "SS" without providing requisite definitional clarity or terminological explication. Scholarly convention necessitates that all abbreviations, particularly those serving as categorical identifiers within tabular presentations, receive proper elucidation. The authors must provide explicit definition of this abbreviation within the table notes to ensure unambiguous interpretation of the tabular content, thus conforming to fundamental principles of scholarly discourse and scientific communication.

Reviewer #2: The data quality in this paper is very poor, and the model has significant flaws, as it fails to consider the key triggering mechanisms of debris flows. The study lacks an in-depth analysis of critical geological and hydrological variables, making the proposed deep learning approach questionable in terms of its applicability to debris flow prediction. The model is entirely data-driven without sufficient justification for its reliability or physical relevance. Furthermore, the discussion section lacks scientific value, as it does not provide a meaningful interpretation of the prediction results. The SHAP-based analysis of variable importance also fails to offer practical disaster mitigation strategies. Overall, this paper lacks both practical and scientific value. Rejection is recommended.

**Do you want your identity to be public for this peer review?** For information about this choice, including consent withdrawal, please see our Privacy Policy

Reviewer #1: No

Reviewer #2: No

---

## [Author Response · Author response to Decision Letter 1]

30 Apr 2025

Response to Editor and Reviewers

Dear editors and reviewers,

Thank you for your letter and for the reviewers' comments concerning our manuscript entitled “An Integrated IKOA-CNN-BiGRU-Attention Framework with SHAP Explainability for High-Precision Debris Flow Hazard Prediction in the Nujiang River Basin, China” (ID: PONE-D-25-14786). Those comments are all valuable and very helpful for revising and improving our paper, as well as providing important guiding significance to our research. We have studied comments carefully and have made corrections which we hope meet with approval. We also highlighted the changes using red fonts in the attached Revised full manuscript file to make it easier for you to check our revision. The principal revisions made within this paper and responses to the reviewers' comments are delineated as follows.

Sincerely Yours,

Dr. Tianlong Wang

Response to Reviewer 1

(Original comments and queries are in blue, while our responses are in black)

[General Comment]

This manuscript presents an integrated framework that combines IKOA-CNN-BiGRU-Attention architecture with SHAP explainability analysis for debris flow hazard prediction in the Nujiang River Basin, China. The research demonstrates innovation in the field of debris flow hazard prediction, particularly in the integration of optimization algorithms, deep learning architectures, and explainability mechanisms. However, some methodological issues should be addressed, particularly regarding the validation of prediction accuracy, reconciliation of variable importance rankings, and justification of analytical choices.

Response:

We thank Reviewer 1 for their recognition of the manuscript’s methodological innovation and for providing nine detailed and constructive comments, which have been instrumental in refining the quality and clarity of our work. In this revision, we have systematically addressed each of the reviewer’s specific points, and all corresponding modifications have been carefully incorporated and are clearly marked in red in the revised manuscript. In particular, Section 6 has been substantially expanded to provide a thorough and transparent explanation of model interpretability, SHAP-based analysis of factor importance, threshold identification, and the mechanistic basis of debris flow initiation, alongside clearer articulation of the operational implications for early warning and risk mitigation strategies. These enhancements are presented in the results and conclusions, with relevant details located in lines 14 to 30, 35 to 147, 387 to 393, 401 to 414, 462 to 632, 674 to 720. The manuscript now clarifies the integrated and complementary roles of comprehensive weighting and SHAP analysis for interpreting variable significance, and it provides a more rigorous justification for key methodological choices, including multicollinearity diagnosis and the hierarchical model architecture. Each of the reviewer’s nine comments, spanning data description, factor selection, interpretation, model comparison, and practical application, has been addressed through targeted clarifications or new analyses as detailed in our point-by-point responses and reflected in the red-marked revisions. We are confident that these comprehensive, transparent, and rigorously implemented revisions significantly enhance the scientific strength, interpretability, and practical relevance of the manuscript. We sincerely thank the reviewer for their guidance, which has been essential to these improvements.

[Comment 1] The introduction's exposition of five paradigms in debris flow prediction research requires structural refinement. The transition between the discussion of the "black box problem" (lines 87-94) and XAI methodologies (lines 95-103) lacks sufficient logical connectivity, thereby undermining the theoretical justification for adopting SHAP methodology in this investigation. I recommend incorporating appropriate transitional discourse to articulate explicitly why SHAP methodology constitutes an optimal approach for addressing the research questions, and to establish a more robust conceptual linkage between the chosen methodology and the research objectives.

Response:

Thank you for this valuable suggestion regarding the structural refinement of the introduction. We fully agree that a robust logical transition between the discussion of the “black box problem” and the adoption of XAI methodologies is fundamental to establishing a solid theoretical foundation for our study. In the revised manuscript, we have addressed this concern by introducing a new paragraph at lines 96–103, which explicitly articulates the three core research gaps in the existing literature. This addition establishes a critical conceptual connection between the inherent limitations of conventional machine learning approaches, particularly their well-documented explainability deficit, and the imperative need for adopting transparent, interpretable methodologies to effectively address persistent challenges in debris flow hazard prediction. Building on this foundation, we have further strengthened the rationale for selecting the SHAP methodology by emphasizing its rigorous mathematical basis in cooperative game theory (line 107), and by citing its demonstrated analytical power in recent earth science applications (lines 108–113). Finally, we have clarified in lines 114–121 how our methodological framework, centered on the integrated use of IKOA-CNN-BiGRU-Attention and SHAP, is purposefully designed to resolve these research gaps and to align directly with our stated research objectives.

[Comment 2] The experimental results presented in Table 3 indicate that the IKOA-CNN-BiGRU-Attention model achieves remarkably low error metrics: RMSE of 2.33×10-6, MAE of 1.51×10-6, and MAPE of 0.006%. Given the inherent complexity and stochasticity of debris flow phenomena, characterized by non-linear processes and substantial data uncertainties, such extraordinarily precise predictions warrant careful scrutiny. The authors should provide a comprehensive explanation of the validation procedures employed to ensure result reliability, discuss potential limitations of the model when deployed in real-world scenarios, and elucidate the practical significance of such high-precision forecasting capabilities for operational debris flow warning systems.

Response:

We thank the reviewer for highlighting the necessity of critically scrutinizing the remarkably low error metrics achieved by our model, taking into account the inherent complexity and uncertainty of debris flow phenomena.

To ensure the reliability of our results, we have incorporated a comprehensive suite of validation procedures. The stability of the IKOA-CNN-BiGRU-Attention model was assessed through fifty independent experimental runs, with results demonstrating exceptional reproducibility and minimal performance fluctuation, as shown by the RMSE of 2.33×10-6 with a standard deviation of 1.51×10-7 (lines 634 to 646, Table 5). Robustness was further evaluated by introducing random noise at levels of five, ten, and twenty percent into the test data, revealing only marginal increases in RMSE, which underscores the model’s resilience to the types of data uncertainty frequently encountered in real-world geomorphological and meteorological measurements (lines 647 to 660, Table 6). Applicability was established via five-fold cross-validation, which confirmed consistent predictive performance across multiple data partitions and verified the model’s generalizability to new geographical contexts (lines 661 to 672, Table 7).

The practical significance of these high-precision capabilities is substantiated in Section 6, where explainability analyses using SHAP provide both interpretable and actionable insights. We have identified quantitative thresholds for key risk factors, such as a critical inflection point for source energy at 3.00×109 J/m2 and a maximum 24-hour rainfall value of 27.01 mm, which can serve as rigorous criteria for operational warning system design (lines 462 to 577). Furthermore, the analysis of factor interactions demonstrates that rainfall only meaningfully increases risk when source energy exceeds approximately 5×10⁹ J/m², thereby supporting the development of hierarchical and context-sensitive early warning protocols. Our assessment of intervention efficacy reveals that rainfall control measures can reduce risk by 58.8 percent, whereas source energy interventions achieve a 25 percent reduction, offering objective guidance for the prioritization of disaster mitigation strategies (lines 578 to 630).

We also recognize that, while the specific threshold values determined in this study are optimized for the Nujiang River Basin, the underlying methodological framework remains transferable and can be regionally calibrated for other settings, as discussed in lines 626 to 630. This recognition of both the contextual specificity and the generalizability of our approach ensures its practical utility for operational systems. Finally, the manuscript now provides explicit recommendations for real-world implementation, including dynamic adjustment of rainfall thresholds based on source energy conditions, integration of temperature and humidity monitoring, and prioritization of interventions according to threshold proximity, all of which are intended to support effective and adaptive risk management in diverse debris flow-prone environments.

[Comment 3] A salient discrepancy exists between the SHAP global explanation analysis (lines 449-479) and the comprehensive weighting results (lines 408-425). Figure 8 identifies potential source energy (X7, 26.98%), catchment area (X8, 17.38%), and main channel length (X11, 9.65%) as the predominant influential variables, whereas Figure 10's SHAP analysis presents a divergent hierarchical ordering, with X1 (maximum 24-hour rainfall) and X2 (annual average rainfall) occupying the second and third positions, respectively. This inconsistency in factor importance ranking necessitates a thorough explication to reconcile these seemingly contradictory findings and to strengthen the credibility of the research conclusions.

Response:

We thank the reviewer for highlighting the apparent discrepancy between the comprehensive weighting results and the SHAP global explanation analysis. This difference does not represent a contradiction but rather reflects the distinct methodological objectives and analytical stages of the two approaches. The comprehensive weighting method, based on the integration of the coefficient of variation, entropy weight, and CRITIC methods using game theory principles (lines 150 to 175), is applied during the pre-modeling stage to assess the intrinsic statistical importance of each factor within the dataset. In contrast, the SHAP analysis provides a post-modeling interpretation by quantifying the marginal contribution of each variable to the model’s output, as determined by the specific nonlinear interactions learned by the neural network (lines 462 to 507). Furthermore, SHAP uniquely accounts for complex conditional feature interactions, as demonstrated in Fig 10(c) and discussed in lines 488 to 499, which the comprehensive weighting approach does not capture.

It is important to note that both methods identify potential source energy (X7) as the most influential variable, thereby providing mutual validation and supporting the dual-threshold conditional triggering mechanism highlighted in lines 703 to 710. The divergence in the ranking of secondary factors, with the comprehensive weighting method emphasizing geographical features (X8, X11) and the SHAP analysis highlighting meteorological and environmental variables (X1, X2, X3, X5, X16), reveals complementary dimensions of debris flow genesis. The comprehensive weighting results suggest that geographical variables serve as the fundamental structural determinants of hazard susceptibility at the catchment scale. In comparison, the SHAP analysis demonstrates that, following the establishment of these geographical preconditions, meteorological and environmental factors act as regulatory modulators and conditional activation triggers. This complementary perspective clarifies the hierarchical causality of debris flow initiation, with geographical features establishing baseline susceptibility, source energy accumulation acting as a necessary precondition, and meteorological conditions providing the critical activation trigger. These insights directly inform hazard management strategies, as elaborated in Section 6.3 (lines 578 to 632), and together strengthen the credibility and interpretability of our research conclusions.

[Comment 4] The SHAP local explanation analysis (lines 480-507) selectively examines only sample points 15 and 37, without providing any methodological rationale for this selection. If these particular sample points were chosen as representative case studies, their geological characteristics and typological representativeness should be explicitly delineated; alternatively, if they were randomly selected, the sampling methodology should be transparently documented to mitigate concerns regarding selection bias. Enhancement of this section will substantially augment the persuasiveness of the SHAP analytical findings.

Response:

We thank the reviewer for identifying the need for greater methodological transparency regarding the selection of sample points in our SHAP local explanation analysis. In response, we have substantially revised Section 6.3 (lines 578 to 632) to provide a clear and rigorous account of our sampling methodology and to ensure the representativeness of the selected cases.

First, the revised manuscript now explicitly states that sample points for SHAP local analysis were chosen using a stratified random sampling approach, as described in lines 598 to 606. This method ensures balanced representation across the full spectrum of predicted hazard levels and mitigates concerns regarding selection bias. By employing stratified sampling, we captured the substantial heterogeneity in localized risk factor compositions, thereby justifying the focus on diverse representative cases rather than arbitrary or isolated examples.

Second, we have clarified the geological and typological representativeness of the selected points by directly linking each sample to its specific geomorphological context. As noted in lines 607 to 625, this approach demonstrates that effective risk management requires location-specific strategies tailored to the predominant risk factors present in each geological setting.

Third, the practical implications of the local SHAP analysis have been strengthened. The manuscript now highlights that locations dominated by meteorological factors would benefit most from enhanced weather-based warning systems, while areas where topographical factors predominate are better suited to physical infrastructure interventions. These distinctions are articulated in lines 607 to 625, providing clear operational guidance informed by the local analysis.

Finally, the local explanation analysis has been fully integrated into our broader analytical framework. The revised text explicitly connects these findings to our precision debris flow risk management strategy (lines 607 to 625), demonstrating how local factor heterogeneity informs actionable interventions within the practical framework proposed. Furthermore, we have expanded the discussion of intervention strategies by recommending a hierarchical management approach, including the classification of areas by source energy potential and tailored mitigation measures for different risk zones (lines 578 to 632).

[Comment 5] The multicollinearity diagnostic procedure (lines 385-394) employs VIF<5 as a threshold criterion, rather than the more conventionally accepted standard of VIF<10 in statistical literature. The authors should provide a theoretical or empirical justification for adopting this more stringent threshold, and should include the original VIF values for the eliminated variables (X10, X14, X17, and X19) to enable readers to independently evaluate the validity of the variable selection process. Such transparency will enhance the methodological rigor and reproducibility of the rese

---

## [Decision Letter · Decision Letter 1]

Dear Dr. Wang,

In addition, I would like to bring to your attention that citing the papers suggested by the reviewers is not mandatory for your revised manuscript. It is entirely up to you whether or not you choose to include the suggested papers in your revised version. The reviewers have provided these suggestions to enhance the quality and credibility of your research, but ultimately, the decision is yours. You have the freedom to decline including any of the suggested papers in your revised manuscript if you feel they are not relevant or do not add value to your study.

Meanwhile, I want to thank the reviewers for their great efforts on your manuscript and you for your submission.

We look forward to receiving your revised manuscript.

Kind regards,

Linwei Li

Academic Editor

PLOS ONE

Reviewers' comments:

Reviewer's Responses to Questions

**Comments to the Author**

Reviewer #1: All comments have been addressed

Reviewer #3: (No Response)

2. Is the manuscript technically sound, and do the data support the conclusions?

Reviewer #1: Yes

Reviewer #3: (No Response)

3. Has the statistical analysis been performed appropriately and rigorously?

Reviewer #1: Yes

Reviewer #3: (No Response)

4. Have the authors made all data underlying the findings in their manuscript fully available?

Reviewer #1: Yes

Reviewer #3: (No Response)

5. Is the manuscript presented in an intelligible fashion and written in standard English?

Reviewer #1: Yes

Reviewer #3: (No Response)

Reviewer #1: The author has revised the manuscript according to the comments. I have no further comments and recommend it for acceptance.

Reviewer #3: This study develops an interpretable deep learning model for predicting debris flow hazards in the Nujiang River Basin, China, leveraging advanced neural network architectures and optimization techniques. The manuscript is clearly an exceptional effort by the authors and the presented model provides a scalable and explainable tool for geohazard prediction, improving risk mitigation strategies in data-scarce mountainous regions. Yet, few adjustments are needed before being considered for publication, and I would be ready to assess the revised version again after applying the following adjustments:

The manuscript is somewhat lengthy. Try reducing redundant debates and keep all ideas short and concise.

Line 65-66: You stated that “Concurrently, multi-attribute decision frameworks provided structured methodologies for integrating heterogeneous hazard factors” add a direct reference utilizing similar frameworks in risk assessment such as:

• Mohseni, U., Jat, P. K., & Siriteja, V. (2025). Multi-criteria analysis-based mapping of the cyclone-induced pluvial flooding in coastal areas of India. DYSONA-Applied Science, 6(2), 309-321. https://doi.org/10.30493/das.2025.490282

Line 91: black-box problem is not well defined, although it represents a corner stone to justify the utilization of explanatory techniques such as SHAP. Try explaining the theory behind it more.

Line 141-147: omit this section

The final part of the introduction should be reserved to summarize the motives of your research and state the aims in light of these motives.

Material and Methodology (and related sections) span 15 pages (7-22). This is an example of a section that can be reduced in size. I understand the inclusive nature of mathematical modelling that you have incorporated into this section; however, the extensive length might render it hard to follow.

Add more details on Fig 5. Such as input sizes, filter sizes, neuron numbers …etc. All the information that can assist rebuilding the (basic) model.

**Do you want your identity to be public for this peer review?** For information about this choice, including consent withdrawal, please see our Privacy Policy

Reviewer #1: No

Reviewer #3: No

---

## [Author Response · Author response to Decision Letter 2]

19 May 2025

Response to Editor and Reviewers

Dear editors and reviewers,

Thank you for your letter and for the reviewers' comments concerning our manuscript entitled “An Integrated IKOA-CNN-BiGRU-Attention Framework with SHAP Explainability for High-Precision Debris Flow Hazard Prediction in the Nujiang River Basin, China” (ID: PONE-D-25-14786R1). Those comments are all valuable and very helpful for revising and improving our paper, as well as providing important guiding significance to our research. We have studied comments carefully and have made corrections which we hope meet with approval. We also highlighted the changes using red fonts in the attached Revised full manuscript file to make it easier for you to check our revision. The principal revisions made within this paper and responses to the reviewers' comments are delineated as follows.

Sincerely Yours,

Dr. Tianlong Wang

Response to Reviewer 1

(Original comments and queries are in blue, while our responses are in black)

[General Comment]

The author has revised the manuscript according to the comments. I have no further comments and recommend it for acceptance.

Response: We sincerely thank Reviewer 1 for taking the time to re-evaluate our manuscript and for acknowledging the improvements made in response to the previous round of comments.

Response to Reviewer 3

(Original comments and queries are in blue, while our responses are in black)

[General Comment] This study develops an interpretable deep learning model for predicting debris flow hazards in the Nujiang River Basin, China, leveraging advanced neural network architectures and optimization techniques. The manuscript is clearly an exceptional effort by the authors and the presented model provides a scalable and explainable tool for geohazard prediction, improving risk mitigation strategies in data-scarce mountainous regions. Yet, few adjustments are needed before being considered for publication, and I would be ready to assess the revised version again after applying the following adjustments:

Response:

We sincerely thank Reviewer 3 for the positive assessment of our manuscript and for recognizing the potential contribution of our work to improving geohazard prediction and risk mitigation in mountainous regions. We appreciate the constructive feedback and have carefully addressed each suggestion to enhance the clarity, conciseness, and technical depth of our manuscript.

[Comment 1] The manuscript is somewhat lengthy. Try reducing redundant debates and keep all ideas short and concise.

Response:

Thank you very much for pointing out this important issue. The reviewer’s suggestion regarding manuscript length and conciseness is greatly appreciated. In response to this valuable comment, the manuscript has been carefully revised with the aim of reducing redundancy and improving clarity and readability. Specifically, revisions have been implemented in several key sections as follows:

(1) In Section 1, theoretical background discussions, research motivations, and objectives have been streamlined. Redundant explanations and peripheral details have been removed, resulting in a more focused and concise presentation of the research context and aims.

(2) In Section 2, lengthy mathematical derivations and detailed algorithm descriptions have been condensed, retaining only essential equations and methodological steps required for reproducibility. This revision significantly reduces the complexity and length of the section.

(3) Section 4 has been revised to succinctly summarize data sources, data processing methods, and multicollinearity diagnostics. Unnecessary details have been removed, enhancing the clarity and conciseness of the section.

(4) Section 6 has been simplified by highlighting key findings and practical implications through concise descriptions and clear visualizations. Excessive analytical discussions have been appropriately condensed.

(5) Section 8 has been carefully edited to succinctly summarize the main contributions and implications of the study. Repetitive statements and non-essential content have been eliminated to ensure conciseness.

Through these targeted revisions, the manuscript length has been reduced from approximately 9,500 words to approximately 8,000 words (excluding references), thereby effectively addressing the reviewer’s valuable comment regarding the manuscript length. The authors sincerely appreciate the reviewer’s constructive recommendation, which has greatly improved the manuscript quality.

[Comment 2] Line 65-66: You stated that “Concurrently, multi-attribute decision frameworks provided structured methodologies for integrating heterogeneous hazard factors” add a direct reference utilizing similar frameworks in risk assessment such as:

• Mohseni, U., Jat, P. K., & Siriteja, V. (2025). Multi-criteria analysis-based mapping of the cyclone-induced pluvial flooding in coastal areas of India. DYSONA-Applied Science, 6(2), 309-321. https://doi.org/10.30493/das.2025.490282

Response:

We have incorporated the suggested citation to strengthen our discussion of multi-attribute decision frameworks in risk assessment. The revised sentence now reads (lines 65-69 in the revised manuscript): “Concurrently, multi-attribute decision frameworks provided structured methodologies for integrating heterogeneous hazard factors [8], with recent innovations applying Analytic Hierarchy Process techniques to susceptibility mapping across diverse geological contexts [9] and multi-criteria analysis approaches to flood risk assessment in coastal regions [10].”

References

8. Onaopemipo Akintola M. Enhancing disaster response and resilience through near-time GIS for flood monitoring and analysis in Niger river basin, nigeria. Int Arch Photogramm Remote Sens Spat Inf Sci. 2024;XLVIII-3–2024: 377–385. doi:10.5194/isprs-archives-XLVIII-3-2024-377-2024

9. Barman J, Das J. Assessing classification system for landslide susceptibility using frequency ratio, analytical hierarchical process and geospatial technology mapping in aizawl district, NE india. Adv Space Res. 2024;74: 1197–1224. doi:10.1016/j.asr.2024.05.007

10. Mohseni U, Jat PK, Siriteja V. Multi-criteria analysis-based mapping of the cyclone-induced pluvial flooding in coastal areas of India. DYSONA - Appl Sci. 2025;6: 309–321. doi:10.30493/das.2025.490282

[Comment 3] Line 91: black-box problem is not well defined, although it represents a corner stone to justify the utilization of explanatory techniques such as SHAP. Try explaining the theory behind it more.

Response:

The reviewer’s comment regarding the insufficient definition of the "black-box problem" is highly relevant. This concept indeed serves as a fundamental justification for employing SHAP-based interpretability approaches. In response, the explanation of the black-box problem has been substantially expanded in the revised manuscript (lines 83-103 in the revised manuscript) to provide a more comprehensive theoretical foundation. The revised text now reads:

“Despite their transformative potential, current ML applications to debris flow hazard assessment exhibit critical epistemological and methodological limitations that constrain their scientific impact and operational utility. First, model performance demonstrates an acute sensitivity to hyperparameter configurations, with suboptimal parameterization inducing overfitting phenomena that compromise generalization to novel geomorphological contexts[14]. Second, conventional optimization algorithms frequently converge prematurely to local extrema, resulting in suboptimal exploration of the high-dimensional parameter space[15]. Third, and most fundamentally, deep learning architectures manifest intrinsic opacity that creates a profound disconnect between predictive capacity and mechanistic understanding[16]. This explainability deficit, frequently characterized as the 'black-box problem,' represents a critical epistemological barrier in hazard forecasting applications. The black-box problem specifically refers to the inherent inability to trace how deep learning models transform input features into output predictions through their complex internal architectures. This opacity manifests through (1) algorithmic complexity, where numerous interconnected neurons and non-linear activation functions obscure input-output relationships; (2) latent feature representation, where models develop abstract internal representations that lack direct physical interpretation; and (3 stochastic learning behavior, where training procedures yield models whose internal configurations cannot be deterministically predicted. Contemporary neural networks, while achieving unprecedented predictive accuracy, typically obscure the contribution of specific variables to prediction outcomes, thereby inhibiting scientific elucidation of causative mechanisms and undermining the implementation of targeted mitigation strategies.”

This expanded explanation articulates the theoretical dimensions of the black-box problem more precisely, providing a clearer foundation for understanding why interpretability methods like SHAP are essential for translating high-performance prediction models into scientifically valuable and actionable insights in debris flow hazard assessment.

[Comment 4] Line 141-147: omit this section

Response:

The reviewer's suggestion to omit lines 141-147 has been implemented in the revised manuscript. This section, which previously detailed the manuscript structure, has been removed entirely. To maintain the valuable visual representation of the research framework, the reference to Fig 1 has been repositioned to the end of the paragraph discussing the methodological framework (line 135). The revised text now reads:

“To address these objectives, this study implements an IKOA-CNN-BiGRU-Attention framework that integrates physics-inspired optimization algorithms with deep learning architectures and explainability mechanisms. This framework combines convolutional networks for spatial feature extraction, bidirectional recurrent units for temporal dynamics modeling, and attention mechanisms for feature importance weighting, specifically designed to capture debris flow triggering mechanisms while providing SHAP-based scientific insights. The framework advances debris flow hazard prediction methodologically, theoretically, analytically, and practically, offering actionable insights for precision disaster risk management. The overall workflow of the study is illustrated in Fig 1.”

[Comment 5] The final part of the introduction should be reserved to summarize the motives of your research and state the aims in light of these motives.

Response:

This valuable suggestion has been implemented by restructuring the final portion of the introduction (lines 124-152) to create a more focused conclusion that explicitly connects research motives to study aims. The revised structure maintains the essential content while reorganizing paragraphs to achieve a more cohesive narrative flow. The restructured final section now reads:

“Despite their transformative potential, current ML applications to debris flow hazard assessment exhibit critical epistemological and methodological limitations that constrain their scientific impact and operational utility. First, model performance demonstrates an acute sensitivity to hyperparameter configurations, with suboptimal parameterization inducing overfitting phenomena that compromise generalization to novel geomorphological contexts[14]. Second, conventional optimization algorithms frequently converge prematurely to local extrema, resulting in suboptimal exploration of the high-dimensional parameter space[15]. Third, and most fundamentally, deep learning architectures manifest intrinsic opacity that creates a profound disconnect between predictive capacity and mechanistic understanding[16]. This explainability deficit, frequently characterized as the 'black-box problem,' represents a critical epistemological barrier in hazard forecasting applications. The black-box problem specifically refers to the inherent inability to trace how deep learning models transform input features into output predictions through their complex internal architectures. This opacity manifests through (1) algorithmic complexity, where numerous interconnected neurons and non-linear activation functions obscure input-output relationships; (2) latent feature representation, where models develop abstract internal representations that lack direct physical interpretation; and (3) stochastic learning behavior, where training procedures yield models whose internal configurations cannot be deterministically predicted. Contemporary neural networks, while achieving unprecedented predictive accuracy, typically obscure the contribution of specific variables to prediction outcomes, thereby inhibiting scientific elucidation of causative mechanisms and undermining the implementation of targeted mitigation strategies.

Three fundamental research gaps persist in contemporary literature that motivate this study: (1) the predominant prioritization of predictive accuracy over model explainability creates an artificial tension between computational performance and theoretical advancement; (2) current methodological frameworks overwhelmingly employ single-algorithm approaches that inadequately capture the complex spatiotemporal characteristics of debris flow triggering conditions; and (3) the systematic integration of advanced optimization techniques with explainable artificial intelligence frameworks remains largely unexplored in geohazard prediction contexts, particularly for multi-parameter debris flow risk assessment. Explainable Artificial Intelligence (XAI) methodologies, particularly the SHapley Additive exPlanations (SHAP) approach, offer promising solutions to these challenges by providing mathematically rigorous attribution mechanisms derived from cooperative game theory principles[17,18]. Despite preliminary applications identifying dominant controlling factors in debris flow research, comprehensive XAI integration within holistic debris flow assessment frameworks remains substantially underdeveloped[19].

Motivated by these research gaps and the potential of XAI approaches, this study aims to develop and validate an integrated deep learning framework that maximizes both predictive accuracy and scientific explainability for debris flow hazard assessment. The specific research objectives include: (1) identifying optimal architectural configurations for capturing complex spatiotemporal characteristics of debris flow triggering factors; (2) quantifying the relative contribution of environmental, geological, and meteorological variables; (3) establishing a theoretical framework connecting physical processes to prediction outcomes through explainable AI; and (4) determining quantitative thresholds governing debris flow risk to support evidence-based intervention strategies.

To address these objectives, this study implements an IKOA-CNN-BiGRU-Attention framework that integrates physics-inspired optimization algorithms with deep learning architectures and explainability mechanisms. This framework combines convolutional networks for spatial feature extraction, bidirectional recurrent units for temporal dynamics modeling, and attention mechanisms for feature importance weighting, specifically designed to capture debris flow triggering mechanisms while providing SHAP-based scientific insights. The framework advances debris flow hazard prediction methodologically, theoretically, analytically, and practically, offering actionable insights for precision disaster risk management. The overall workflow of the study is illustrated in Fig 1.” (lines 83-135)

This restructured conclusion more effectively summarizes the research motives and explicitly connects them to the study aims, creating a stronger narrative arc that guides readers from identified problems to proposed solutions, as suggested by the reviewer.

[Comment 6] Material and Methodology (and related sections) span 15 pages (7-22). This is an example of a section that can be reduced in size. I understand the inclusive nature of mathematical modelling that you have incorporated into this section; however, the extensive lengt

---

## [Decision Letter · Decision Letter 2]

An Integrated IKOA-CNN-BiGRU-Attention Framework with SHAP Explainability for High-Precision Debris Flow Hazard Prediction in the Nujiang River Basin, China

PONE-D-25-14786R2

Dear Dr. Wang,

We’re pleased to inform you that your manuscript has been judged scientifically suitable for publication and will be formally accepted for publication once it meets all outstanding technical requirements.

Kind regards,

Linwei Li

Academic Editor

PLOS ONE

Additional Editor Comments (optional):

None

Reviewers' comments:

Reviewer's Responses to Questions

**Comments to the Author**

Reviewer #3: All comments have been addressed

2. Is the manuscript technically sound, and do the data support the conclusions?

Reviewer #3: Yes

3. Has the statistical analysis been performed appropriately and rigorously?

Reviewer #3: Yes

4. Have the authors made all data underlying the findings in their manuscript fully available?

Reviewer #3: Yes

5. Is the manuscript presented in an intelligible fashion and written in standard English?

Reviewer #3: Yes

Reviewer #3: The authors have addressed all my comments. The manuscript is ready for publication

The authors have addressed all my comments. The manuscript is ready for publication

The authors have addressed all my comments. The manuscript is ready for publication

**Do you want your identity to be public for this peer review?** For information about this choice, including consent withdrawal, please see our Privacy Policy

Reviewer #3: No

---

## [Editor Report · Acceptance letter]

PONE-D-25-14786R2

PLOS ONE

Dear Dr. Wang,

I'm pleased to inform you that your manuscript has been deemed suitable for publication in PLOS ONE. Congratulations! Your manuscript is now being handed over to our production team.

Kind regards,

on behalf of

Dr. Linwei Li

Academic Editor

PLOS ONE